

# TERMS OF USE

All the panels in this comic book are licensed CC BY-NC-ND 4.0. Please refer to the license page for details on how you can use this artwork.

**TL;DR**: Feel free to use panels/groups of panels in your presentations/articles, as long as you
1. Provide the proper citation
2. Do not make modifications to the individual panels themselves

## Cite as:

Falaah Arif Khan, Eleni Manis, and Julia Stoyanovich. "Fairness and Friends". *Data, Responsibly Comics*, Volume 2 (2021)
https://dataresponsibly.github.io/comics/vol2/fairness_en.pdf

## Contact:

Please direct any queries about using elements from this comic to themachinelearnist@gmail.com and cc stoyanovich@nyu.edu

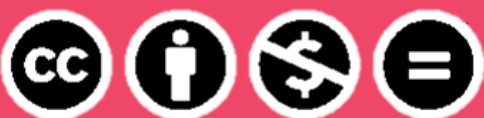

# ACCESSIBILITY STATEMENT

The purpose of scientific publication is the presentation of ideas and dissemination of findings. In the course of our (ongoing) work on creating a comic series about Responsible AI, we have found that relatable cartoons and visual humor are a rich but underappreciated source of clarity and accessibility that enable effective communication to a broad audience. Comic books are a particularly prescient medium for literature reviews and critical surveys, and for bridging insights from different disciplines such as philosophy, law, sociology, and computer science. Given the inherently interdisciplinary nature of machine learning, we see comics and other technical artwork as a promising new medium of scholarship. We hope to demonstrate their utility through our work and to popularize their adoption more broadly in the scientific community.

We care deeply about making our comics as digitally accessible as possible. Towards this end, we have taken the following measures:

1.  We've chosen a typeface that was developed specially for dyslexic readers. All of the major text in the comic is in the "Open Dyslexic" font.

2.  The comic book is fully alt-texted and can be read entirely using a screen reader. We are also releasing a complete transcript of the comic book, including all of the text and image descriptions.

3.  We will be translating the comic into different languages to cater to speakers of languages other than English, as we have done with previous volumes of the Data, Responsibly comic series.

We would like to  thank Amy Hurst and Chancey Fleet for guiding us on the Accessibility front.

Please feel free to reach out to us if you have any recommendations on how we can further improve the accessibility of our comics.

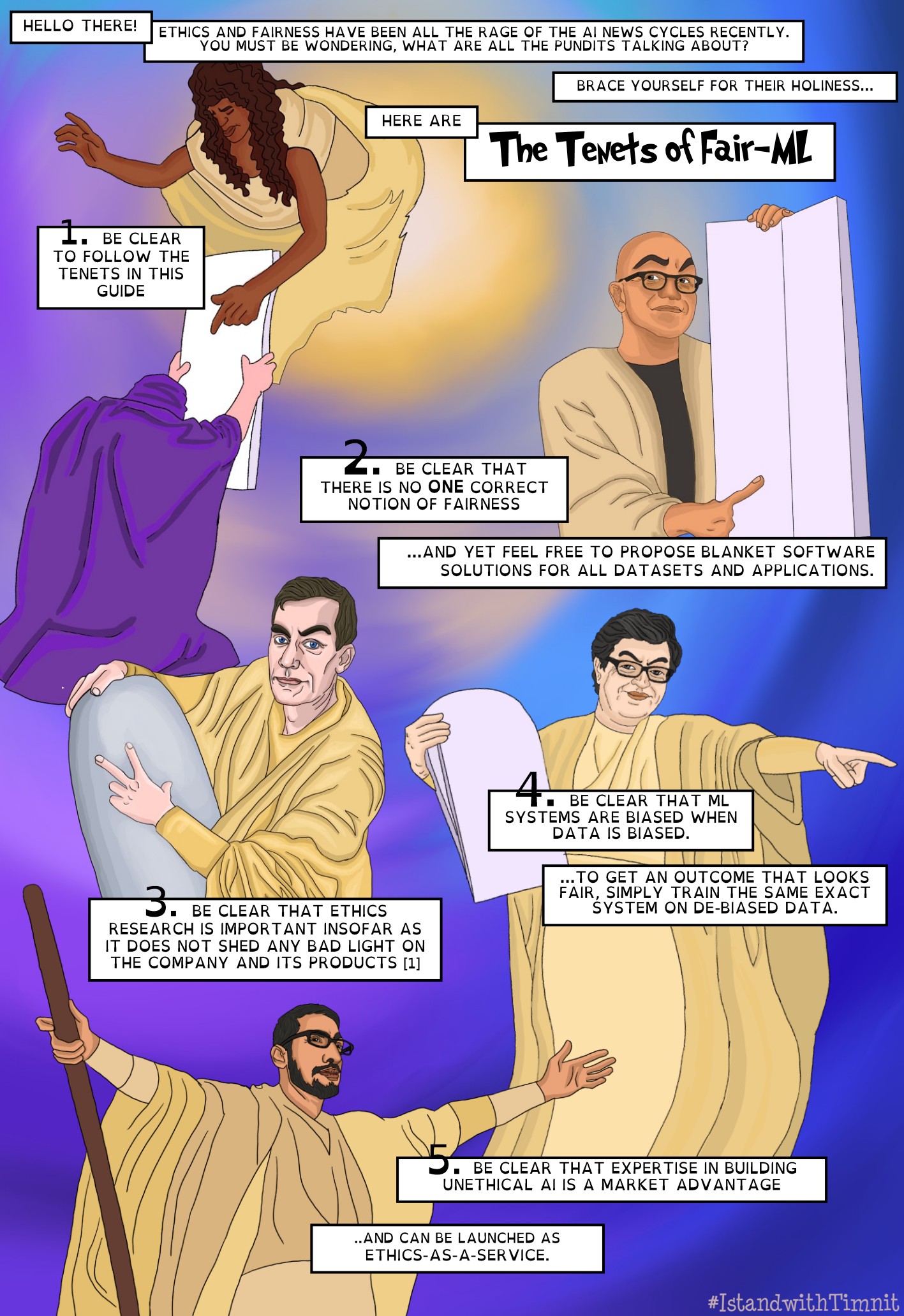

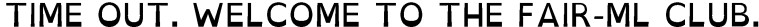

TIME OUT. WELCOME TO THE FAIR-ML CLUB.

**THERE'S ONLY ONE TENET OF FAIR-ML AND IT'S THAT THERE ARE NO TENETS OF FAIR-ML**

FAIRNESS IS **NOT** A TECHNICAL OR STATISTICAL CONCEPT AND THERE CAN NEVER BE A TOOL OR SOFTWARE THAT CAN FULLY 'DE-BIAS' YOUR DATA OR MAKE YOUR MODEL 'FAIR'.

FAIRNESS IS AN ETHICAL CONCEPT, AND A CONTESTED ONE AT THAT. AT BEST, WE CAN SELECT SOME IDEAL OF WHAT IT MEANS TO BE 'FAIR' AND THEN MAKE PROGRESS TOWARDS SATISFYING IT IN OUR PARTICULAR SETTING.

LET'S BACK UP FURTHER, SHALL WE? WHAT ARE WE EVEN TRYING TO MAKE 'FAIR' ? WHAT ARE ALGORITHMS AND WHEN ARE THEY BIASED?

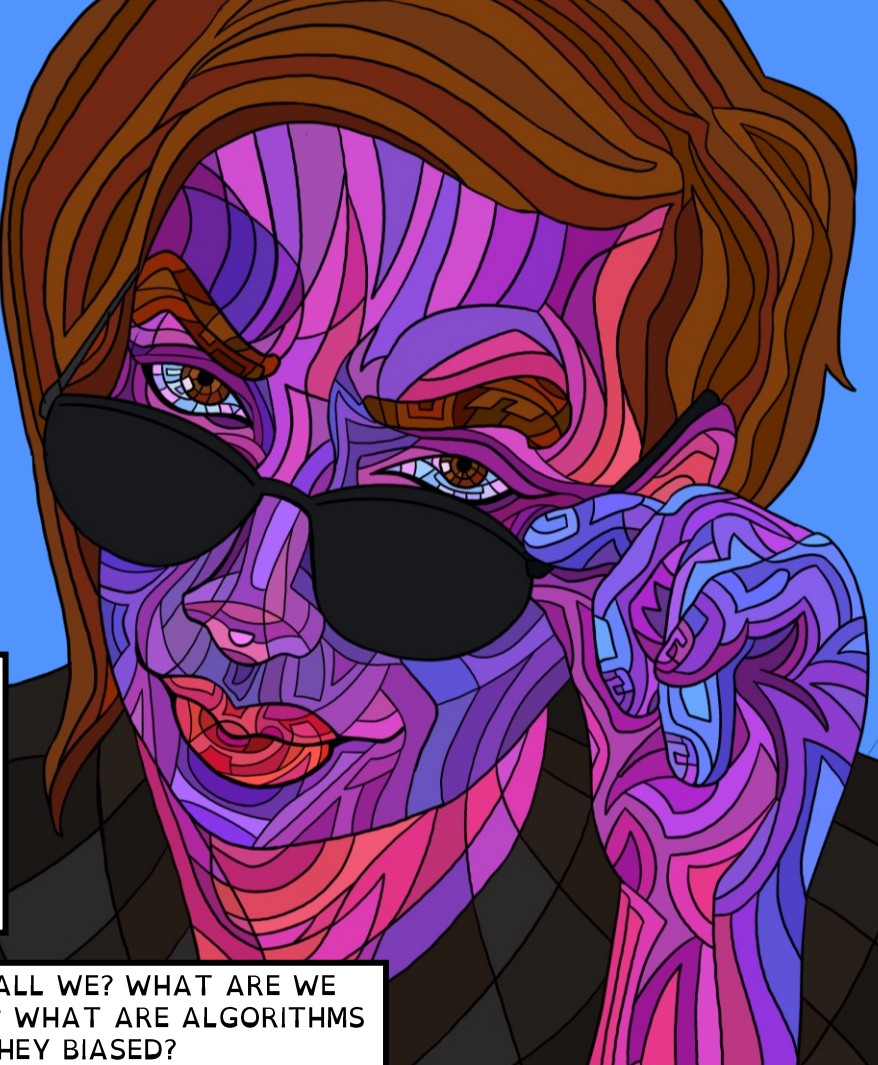

# WHAT IS AN ALGORITHM?

HERE'S A THROWBACK TO THE PREHISTORIC DAYS OF EARLY 2020. REMEMBER THE HOBBY THAT MANY OF US ATTEMPTED TO MASTER - WITH MIXED RESULTS - DURING THE PANDEMIC LOCKDOWN?

## BAKING!

THE RECIPE IS THE ALGORITHM: IT LISTS THE INGREDIENTS AND THEIR PROPORTIONS, AND THE STEPS TO TAKE TO TRANSFORM THEM INTO A SCRUMPTIOUS LOAF.

AKIN TO HOW WE EACH HAVE OUR OWN COOKING STYLES, ALGORITHMS ARE OF DIFFERENT TYPES...

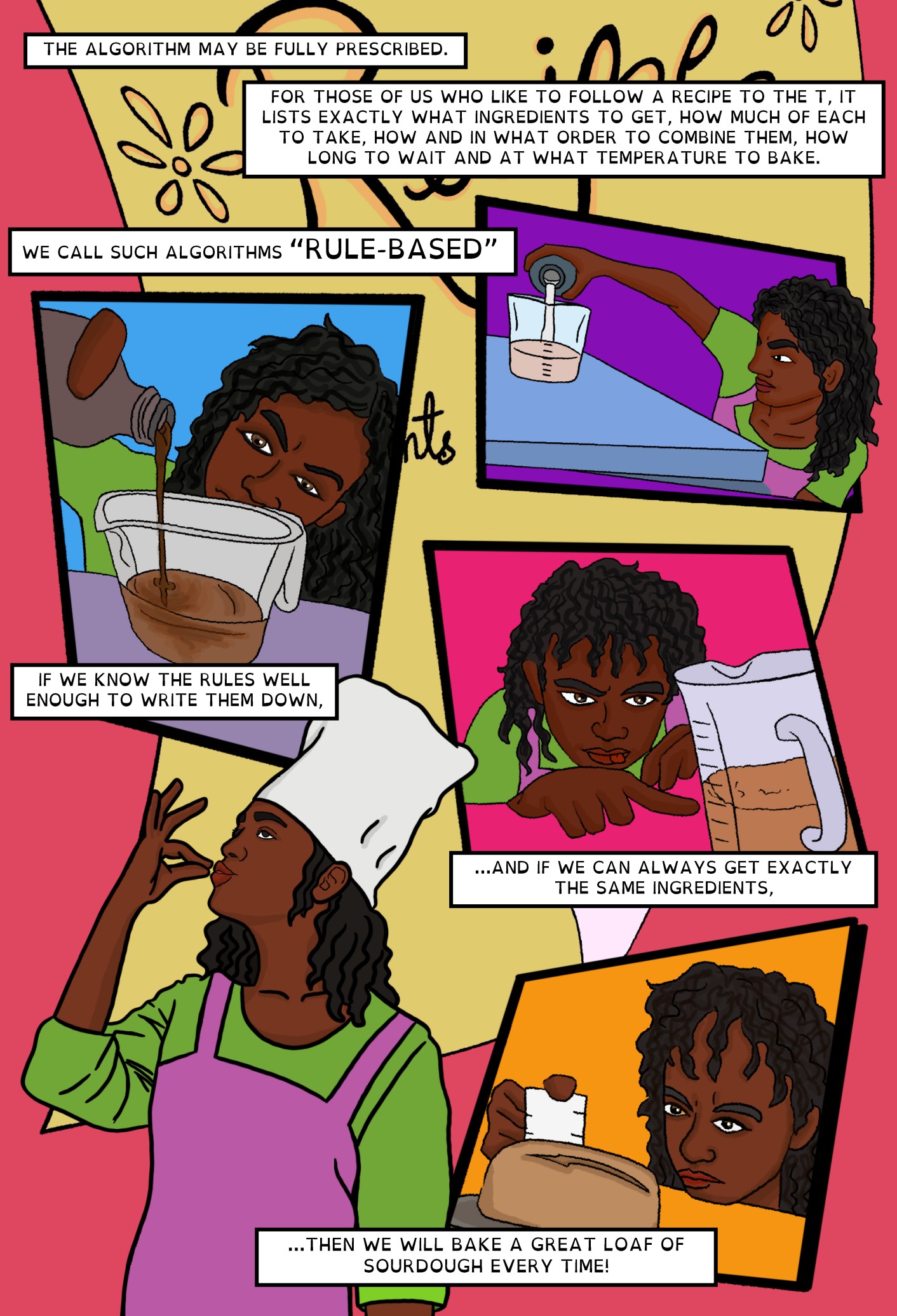

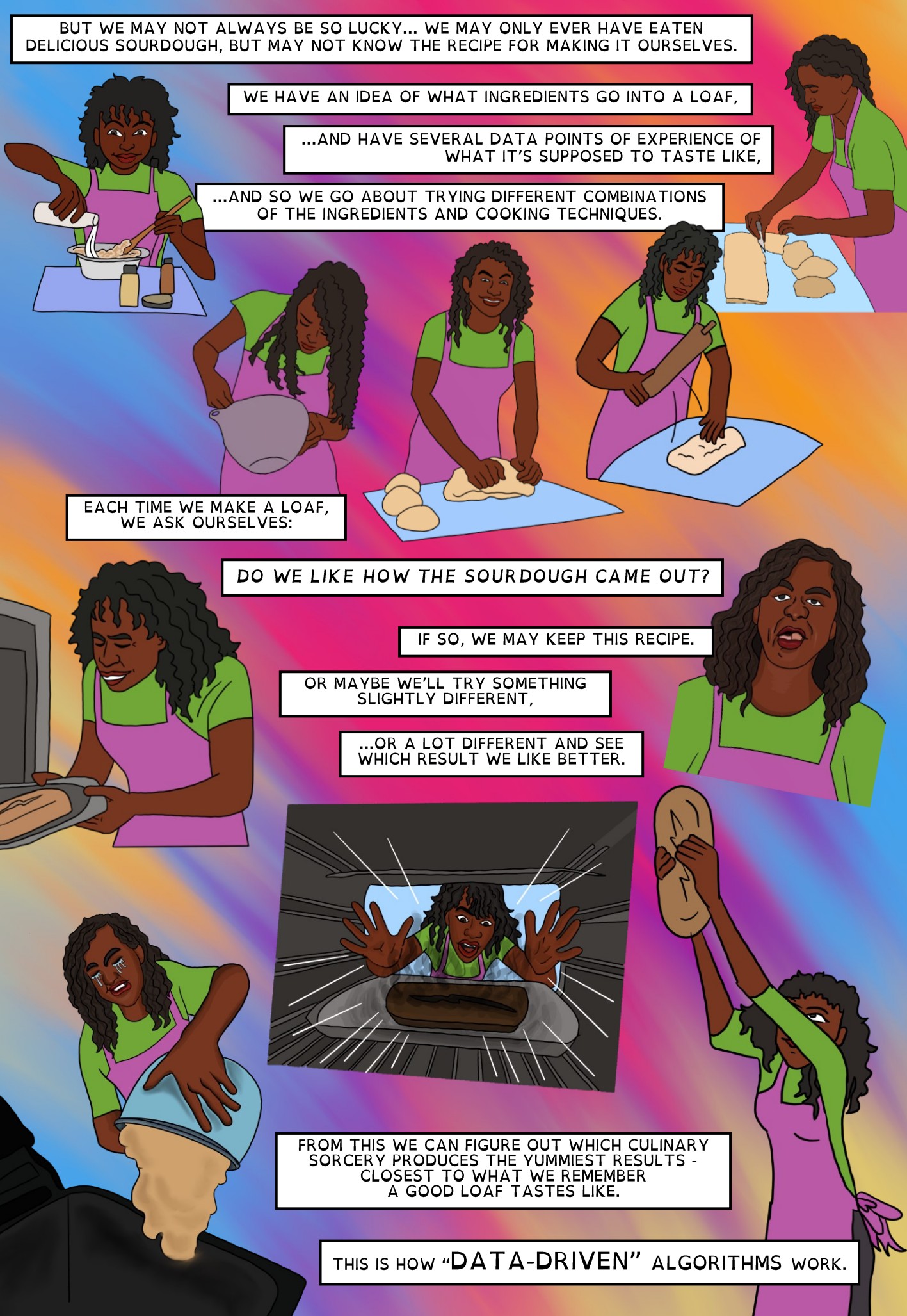

# THE RECIPE IS THE ALGORITHM, NOW WHAT ABOUT **THE DATA?**

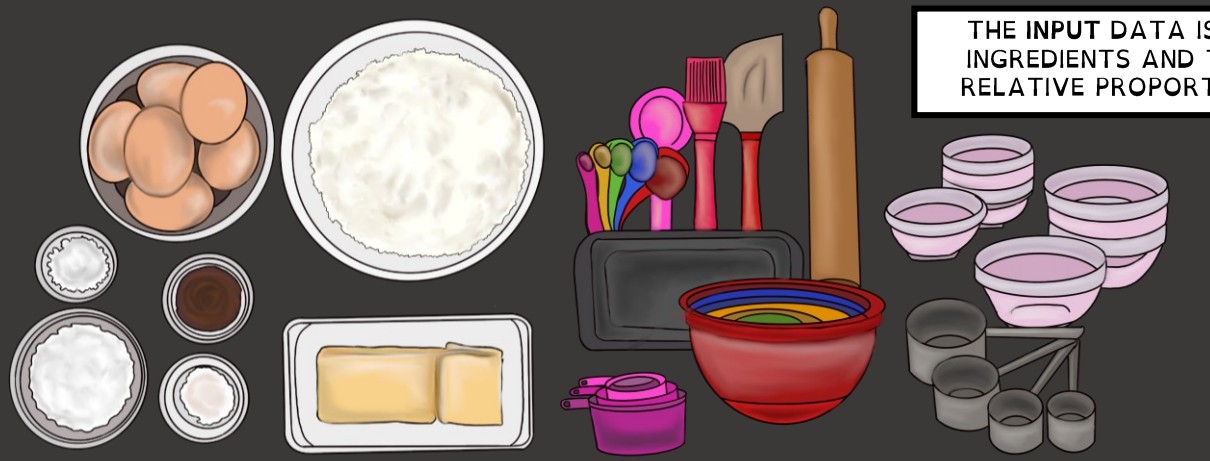

THE **INPUT** DATA IS THE INGREDIENTS AND THEIR RELATIVE PROPORTIONS.

ANOTHER FORM OF DATA IS THE PARAMETER SETTINGS OF YOUR COOKING EQUIPMENT SUCH AS OVEN TEMPERATURE OR WAIT TIMES.

THEY ARE THE KNOBS YOU CAN TURN TO ADJUST THE RECIPE.

THEN THERE'S DATA THAT DESCRIBES THE **OUTPUT**: THAT SCRUMPTIOUS SOURDOUGH THAT WE REMEMBER DEMOLISHING AND ARE HOPING TO BAKE OURSELVES.

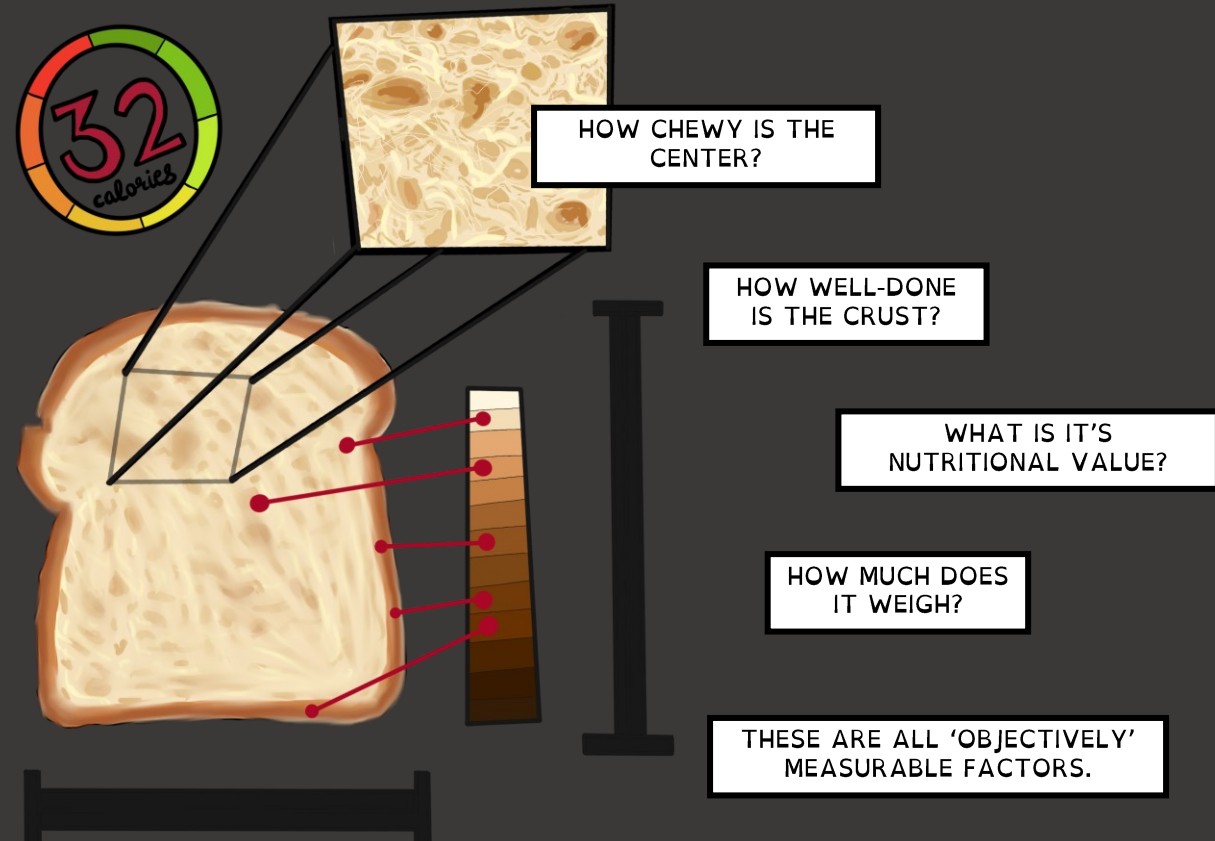

HOW CHEWY IS THE CENTER?

HOW WELL-DONE IS THE CRUST?

WHAT IS IT'S NUTRITIONAL VALUE?

HOW MUCH DOES IT WEIGH?

THESE ARE ALL 'OBJECTIVELY' MEASURABLE FACTORS.

THE FINAL KIND OF DATA IS OUR **REACTION TO THE OUTPUT**

IS IT TASTY?

DOES THE LOAF MEET OUR EXPECTATIONS?

THESE FACTORS BOIL DOWN TO PERSONAL PREFERENCE AND, MORE OFTEN THAN NOT, ARE MORE IMPORTANT THAN THE NUMERICALLY QUANTIFIABLE PROPERTIES OF THE OUTPUT.

# WHAT ABOUT **DECISIONS?**

IN THE PROCESS WE DESCRIBED, IN THE COURSE OF EXECUTION OF THE ALGORITHM, WE ARE FACED WITH SEVERAL DECISIONS.

DOES THE DOUGH LOOK GOOD ENOUGH TO PUT INTO THE OVEN?

HAS THE LOAF RISEN ENOUGH AND SHALL WE TAKE IT OUT OF THE OVEN?

IS THE RESULT INSTAGRAM-WORTHY?

ARE WE GIVING IT A THUMBS UP OR A THUMBS DOWN?

A MORE CONSEQUENTIAL DECISION IS - NOW THAT WE'VE TRIED A BUNCH OF RECIPES, WHICH WILL WE CONSIDER A SUCCESS?

WILL WE SAY THAT IT'S MORE IMPORTANT TO HAVE AN APPETIZING-LOOKING LOAF OR ONE THAT CONSISTENTLY COMES OUT CHEWY ON THE INSIDE AND CRUSTY ON THE OUTSIDE?

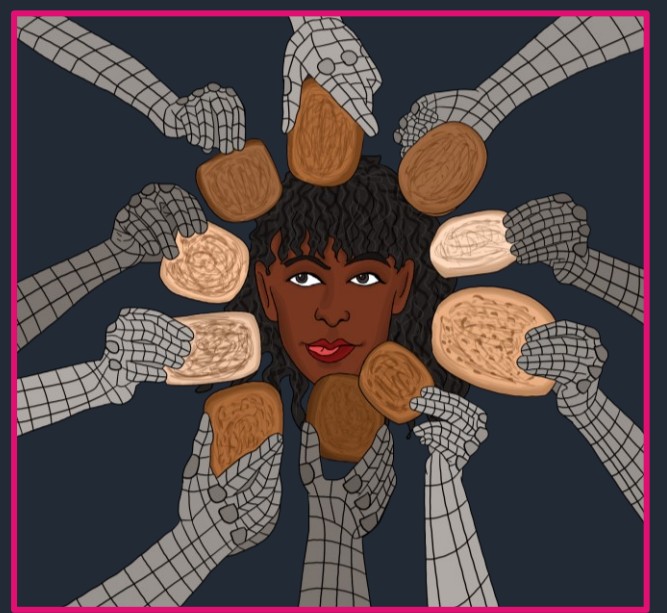 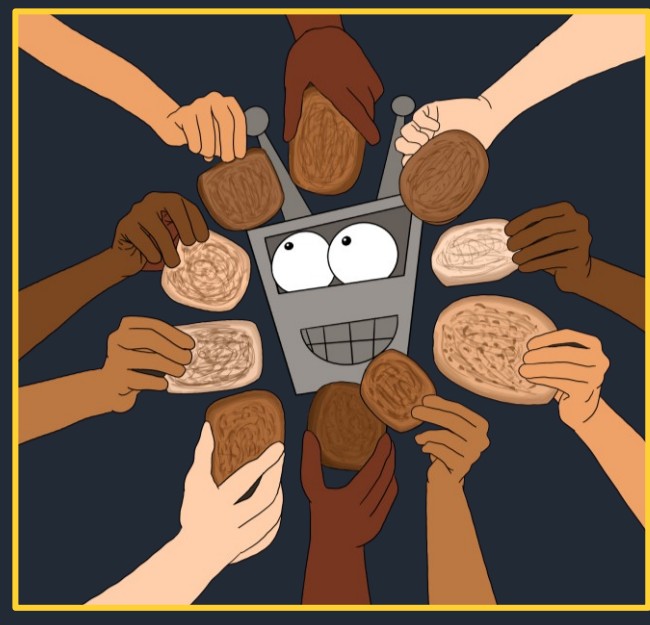

WILL WE DECIDE TO ALWAYS - OR NEVER - USE SOME SPECIFIC INGREDIENTS OR COOKING TECHNIQUES?

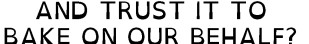

# WHAT IS AN ADS?

SO, AN ALGORITHM IS A RECIPE. THEN, WHAT IS AN AUTOMATED DECISION SYSTEM (ADS) ? IS IT LIKE A SELF-BAKING OVEN?

EASY THERE, MUSK-ETEER.

WE DON'T REALLY HAVE A CONSENSUS ON WHAT AN ADS ACTUALLY IS (OR ISN'T).

THE LAW SEEMS TO HAVE TAKEN A PAGE OUT OF THE 'PAULA ABDUL PLAYBOOK OF JUDGING', GOING OVERLY LENIENT AND VAGUE IN ITS DEFINITION.

NEW YORK CITY'S LOCAL LAW 49 DEFINES AN ADS AS *"COMPUTERIZED IMPLEMENTATIONS OF ALGORITHMS, INCLUDING THOSE DERIVED FROM MACHINE LEARNING OR OTHER DATA PROCESSING OR ARTIFICIAL INTELLIGENCE TECHNIQUES, WHICH ARE USED TO MAKE OR ASSIST IN MAKING DECISIONS."* [2]

USING THIS DEFINITION, ONE COULD ARGUE THAT SPREADSHEETS OR EVEN INTERNET SEARCHES COULD BE ADS, BECAUSE THEY ARE, IN FACT, COMPUTERIZED AND DO, IN FACT, GUIDE DECISION-MAKING. [3]

A PRECISE DEFINITION WILL BE CRUCIAL FOR THE EFFICACY OF ANY ATTEMPT AT REGULATING THESE SYSTEMS. AN ALTERNATE APPROACH WOULD BE TO DEFINE ADS BY EXTENSION. [4]

## SO YOU THINK YOU'RE AN ADS?

DO YOU:

1. PROCESS DATA ABOUT PEOPLE

2. ASSIST - EITHER IN COMBINATION WITH HUMAN DECISION MAKING OR AUTONOMOUSLY - IN MAKING CONSEQUENTIAL DECISIONS THAT IMPACT PEOPLE'S LIVES.

ADDITIONALLY, WE WOULD LIKE IT IF YOU WOULD:

3. HAVE A SPECIFIC, STATED GOAL OF IMPROVING AND PROMOTING EQUALITY AND EFFICIENCY. AT THE VERY LEAST, YOU MUST NOT HINDER EQUITABLE ACCESS TO OPPORTUNITIES

4. BE PUBLICLY DISCLOSED AND SUBJECT TO LEGAL AUDITS.

IS A FORMULA IN A SPREADSHEET AN ADS? PERHAPS – DEPENDS ON WHAT IT'S USED FOR!

IS AN AUTOMATED HIRING TOOL? DEFINITELY.

BUT IS A CALCULATOR AN ADS? NO!

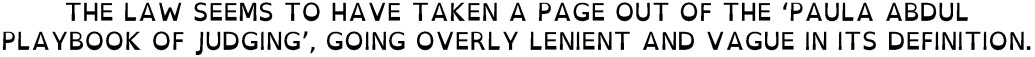

# ALL ABOUT THAT BIAS...

WITH THAT IN MIND, NOW LET'S LOOK AT WHAT WE MEAN BY BIAS IN AN ADS AND HOW IT ARISES. [5]

IN THE CONTEXT OF DATA-DRIVEN SYSTEMS, BIASES ARE 'HARMFUL' ASSOCIATIONS PICKED UP BY THE ALGORITHM - EITHER FROM THE DATA ITSELF, OR FROM HOW THE ALGORITHM IS DESIGNED, OR FROM THE OBJECTIVES THAT WE SPECIFIED FOR IT, OR FROM HOW WE USE IT.

SYSTEMATIC DISCRIMINATION BY AN ALGORITHM IS TERMED 'BIAS'.

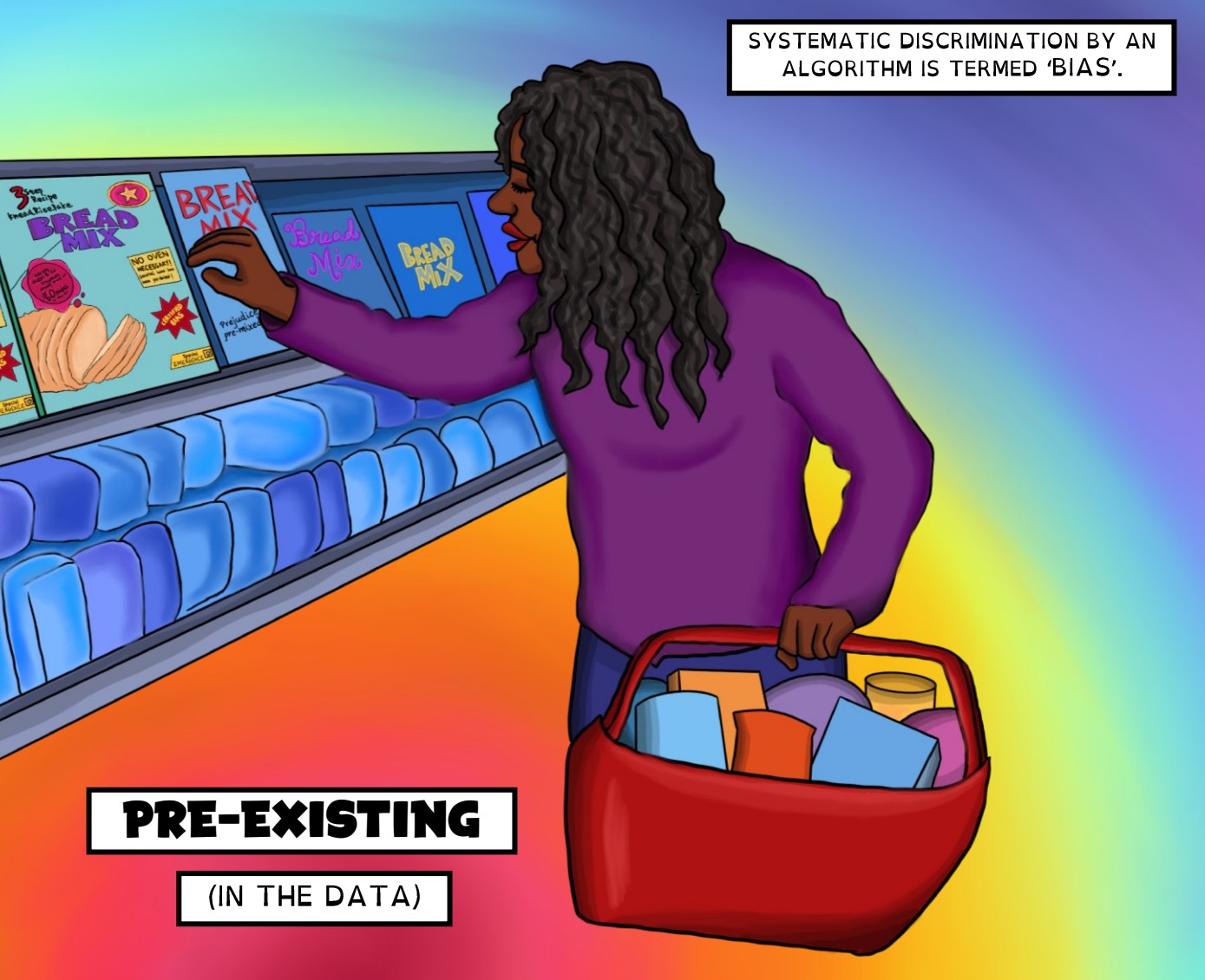

## PRE-EXISTING

(IN THE DATA)

PRE-EXISTING BIASES EXIST IN SOCIETY AND COME 'PRE-BAKED' INTO THE MODEL AS A RESULT OF THE UNDERLYING DISCRIMINATORY SYSTEM THAT THE DATA WAS GENERATED FROM.

THESE WOULD BE THE FLAVOR NOTES THAT WILL SEEP INTO YOUR BREAD IF YOU DON'T PRIORITIZE THE PURITY/FRESHNESS OF YOUR INGREDIENTS OR IF YOU DECIDE TO USE PREMIXED OFF-THE-SHELF BATTER.

A NOTORIOUS EXAMPLE IS THE GENDER AND RACIAL STEREOTYPES THAT LANGUAGE MODELS PICK UP WHEN TRAINED ON DATA FROM SOCIAL MEDIA PLATFORMS.

# TECHNICAL

TECHNICAL BIASES ARE THOSE IMPERFECTIONS THAT WILL SEEP INTO YOUR BREAD IF YOU USE THE WRONG EQUIPMENT.

THINK ABOUT WHAT WOULD HAPPEN IF YOUR OVEN TEMPERATURE IS MISCALIBRATED

OR IF YOUR BAKING EQUIPMENT IS THE WRONG SIZE.

IN THE CONTEXT OF ALGORITHMS, THESE INCLUDE HARDWARE LIMITATIONS, INCORRECT CHOICES OF REPRESENTATION AND STRONG MODELING ASSUMPTIONS THAT ARE NOT SATISFIED IN THE REAL WORLD.

# EMERGENT

(DUE TO DECISIONS)

THE PATTERNS THAT EMERGE AS A RESULT OF YOUR BAKING COMPRISE 'EMERGENT' BIAS.

WHAT IF YOU BECOME SUCH A MAESTRO AT BAKING THAT YOU INADVERTENTLY MAKE BREAD A STEADY PART OF YOUR DIET!

OR MAKE IT SO OFTEN, THAT YOU TURN EVERYONE AROUND YOU OFF THE THOUGHT OF EVER EATING ANOTHER SLICE!

OR THINK ABOUT HOW YOUR IDEA OF 'WHAT BREAD SHOULD TASTE LIKE' IS SHAPED BY THE POPULARITY OF PRODUCTS LIKE 'WONDER BREAD'.

# DATA IS A MIRROR REFLECTION OF THE WORLD. [4]

ALL WE HAVE IS A DISTORTED (BIASED) REFLECTION.

WITHOUT KNOWLEDGE OR ASSUMPTIONS ABOUT THE PROPERTIES OF THE MIRROR AND OF THE WORLD IT REFLECTS, WE CANNOT KNOW WHETHER WE ARE LOOKING AT A DISTORTED REFLECTION OF A PERFECT WORLD OR A PERFECT REFLECTION OF A DISTORTED WORLD OR WHETHER THESE DISTORTIONS COMPOUND. [6]

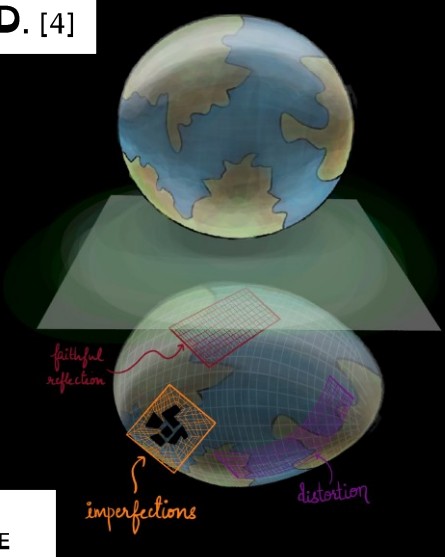

## WHAT IS ALGORITHMIC FAIRNESS?

ALGORITHMIC FAIRNESS IS THE CORRECTIVE LENS THAT WE WEAR IN ORDER TO SEE THE WORLD CLOSER TO WHAT WE WANT IT TO LOOK LIKE THAN WHAT IT ACTUALLY IS.

CORRECTIVE LENSES ARE TAILORED TO THE WEARER AND, SIMILARLY, DIFFERENT INDIVIDUALS JUDGE DIFFERENT FAIRNESS IDEALS TO MATTER, FOR DIFFERENT REASONS.

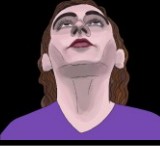

BASED ON OUR WORLDVIEW (BELIEFS ABOUT WHAT THE IDEAL WORLD SHOULD LOOK LIKE), WE APPLY CORRECTIVE MEASURES IN THE FORM OF DIFFERENT STATISTICAL MEASURES OF 'FAIRNESS'.

HOWEVER, WEARING THESE LENSES ONLY CHANGES HOW WE VIEW THE REFLECTION - IT DOES NOT AND CANNOT FIX DISTORTIONS IN THE MIRROR OR FIX DISTORTIONS IN THE WORLD.

UNLESS SUCH FIXES ARE SUPPLEMENTED BY SYSTEMIC CHANGE, WE CAN QUICKLY CONFUSE THE WORLD SEEN THROUGH ROSE-COLORED GLASSES WITH THE REAL WORLD.

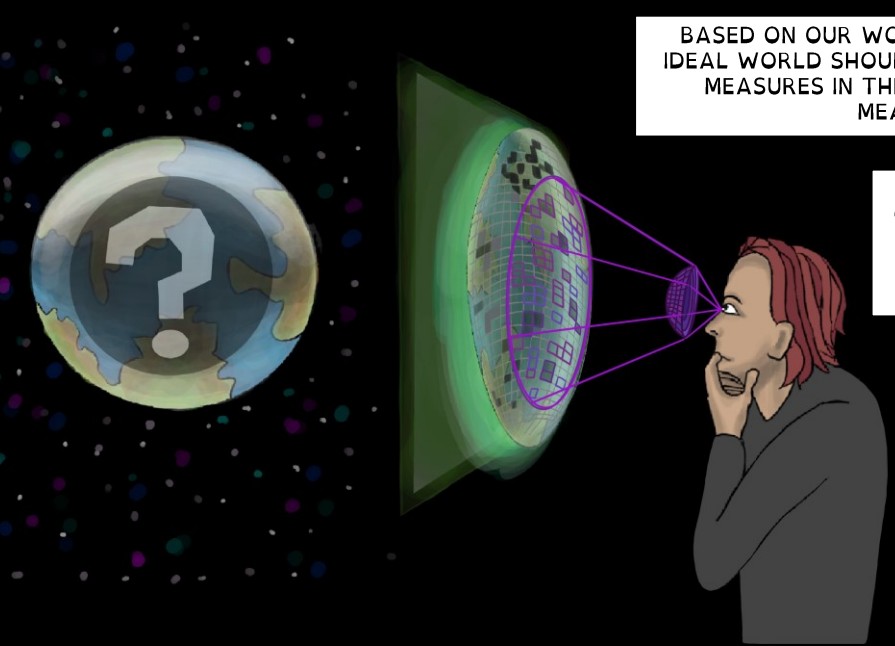

ALGORITHMIC DECISIONS ARE MAPPINGS BETWEEN THREE 'SPACES', NAMELY - THE CONSTRUCT SPACE (THE REAL WORLD), THE OBSERVED SPACE (THE REFLECTION) AND THE DECISION SPACE (THE OUTCOMES OR ALLOCATIONS). [7]

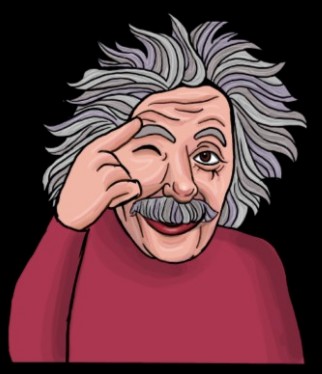

"INTELLIGENCE" IS THE CONSTRUCT.

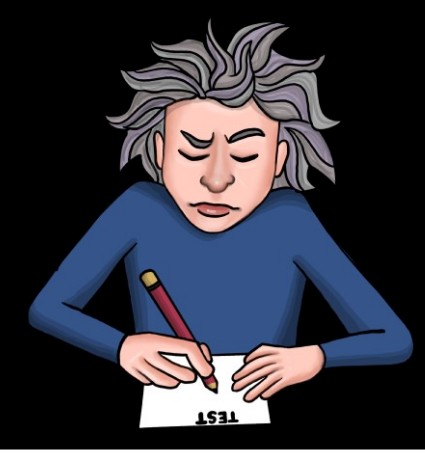

TEST SCORES ARE THE OBSERVATIONS THAT WE ARE ACTUALLY ABLE TO MEASURE.

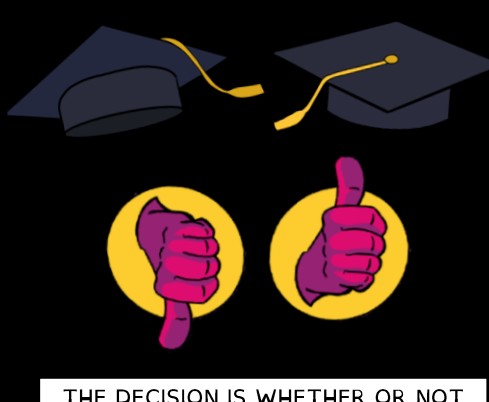

THE DECISION IS WHETHER OR NOT TO CERTIFY ONE'S INTELLECTUAL ABILITY BY CONFERRING UPON THEM A DIPLOMA

IN A PERFECT WORLD, WHERE THERE IS NEITHER A DISTORTION IN THE WORLD NOR IN THE REFLECTION, OUR CONSTRUCTS AND OUR OBSERVATIONS WOULD BE THE SAME.

IN REALITY, THE CONSTRUCT SPACE IS UNOBSERVABLE AND SO WE NEED TO MAKE ASSUMPTIONS ABOUT ITS NATURE AND ABOUT THE MAPPING FROM CONSTRUCT TO OBSERVATION. THESE ASSUMPTIONS COLOR OUR JUDGMENTS ABOUT WHETHER ALLOCATIONS OF BENEFITS ARE 'FAIR' (BY SOME SPECIFIC NOTION).

DIFFERENT WORLDVIEWS AFFECT OUR INTUITIONS ABOUT 'FAIRNESS'. [7]

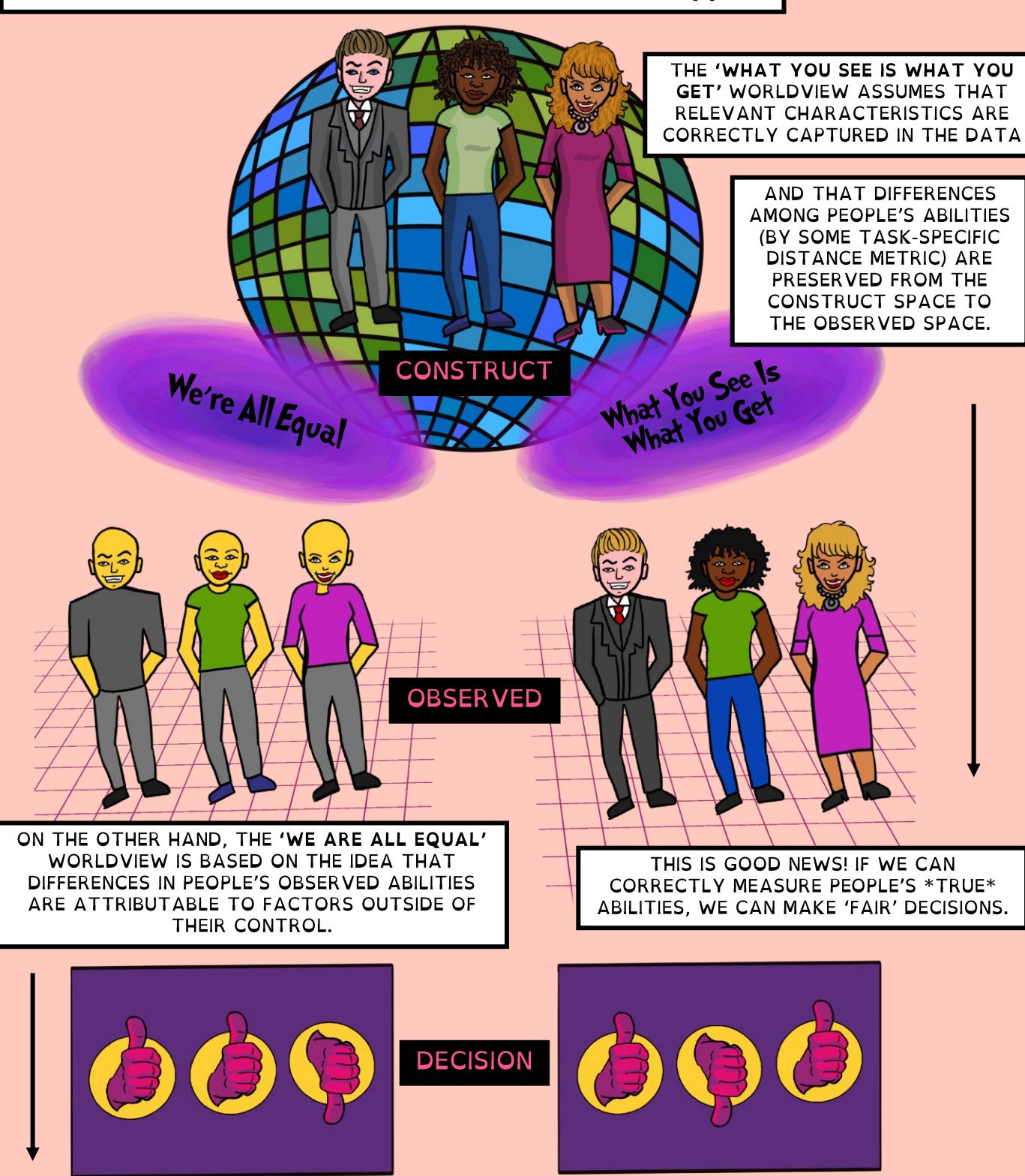

THE 'WHAT YOU SEE IS WHAT YOU GET' WORLDVIEW ASSUMES THAT RELEVANT CHARACTERISTICS ARE CORRECTLY CAPTURED IN THE DATA

AND THAT DIFFERENCES AMONG PEOPLE'S ABILITIES (BY SOME TASK-SPECIFIC DISTANCE METRIC) ARE PRESERVED FROM THE CONSTRUCT SPACE TO THE OBSERVED SPACE.

CONSTRUCT

We're All Equal

What You See Is What You Get

OBSERVED

ON THE OTHER HAND, THE 'WE ARE ALL EQUAL' WORLDVIEW IS BASED ON THE IDEA THAT DIFFERENCES IN PEOPLE'S OBSERVED ABILITIES ARE ATTRIBUTABLE TO FACTORS OUTSIDE OF THEIR CONTROL.

THIS IS GOOD NEWS! IF WE CAN CORRECTLY MEASURE PEOPLE'S *TRUE* ABILITIES, WE CAN MAKE 'FAIR' DECISIONS.

DECISION

IN SO FAR THAT PEOPLE'S ABILITIES CAN BE MEASURED IN A MANNER THAT IS INDEPENDENT OF THEIR PROTECTED CHARACTERISTICS SUCH AS SEX AND RACE, WE CAN MAKE 'FAIR' DECISIONS.

# INDIVIDUAL V/S GROUP

INDIVIDUAL FAIRNESS ADVOCATES THAT 'SIMILAR INDIVIDUALS MUST BE TREATED SIMILARLY'. [8]

MATHEMATICALLY, IF THE DISTANCE BETWEEN TWO PEOPLE, BASED ON SOME TASK-RELEVANT METRIC, IS SMALL, THEN THEY SHOULD BOTH BE ALLOCATED THE SAME OUTCOME.

THE "WHAT YOU SEE IS WHAT YOU GET" WORLDVIEW TRACKS INDIVIDUAL FAIRNESS INSOFAR THAT IT WILL OBJECT TO TWO INDIVIDUALS WHO ARE *TRULY* SIMILAR IN THE CONSTRUCT SPACE, TO APPEAR TO BE DISSIMILAR IN THE OBSERVED SPACE.

GROUP FAIRNESS TRIES TO ENSURE SOME NOTION OF PARITY IN OUTCOMES FOR MEMBERS OF DIFFERENT PROTECTED GROUPS.

HOWEVER, THE CONVERSE NEED NOT BE TRUE – PEOPLE WHO ARE *TRULY* DISSIMILAR IN THE CONSTRUCT SPACE CAN END UP LOOKING SIMILAR IN THE OBSERVED SPACE.

MATHEMATICALLY, WE WOULD AIM TO EQUALIZE SOME STATISTICAL MEASURE - SUCH AS POSITIVE OUTCOMES, ERROR RATES OR FALSE POSITIVE/FALSE NEGATIVE RATES - ACROSS GROUPS.

THINK OF IT AS TWO DIFFERENT COACHING STYLES –

ARE YOU THE DOUG COLLINS OF THE '86-'88 BULLS, DESIGNING YOUR ENTIRE OFFENSE AROUND YOUR MOST TALENTED PLAYER - EAGER TO SEE HIM EARN HIS PLACE AMONG THE ALL-TIME GREATS?

OR ARE YOU THE PHIL JACKSON OF THE BULLS, IDENTIFYING THE DIFFERENT STRENGTHS OF DIFFERENT PLAYERS AND ORGANIZING THE TRIANGLE OFFENSE TO PERFECTION,

...THEREBY TAKING THE BULLS – LED BY THE INIMITABLE JORDAN, OF COURSE - TO THEIR FIRST CHAMPIONSHIP VICTORY.

IN PRINCIPLE, INDIVIDUAL AND GROUP FAIRNESS NEED NOT BE INCOMPATIBLE [9] – YOU CAN PULL OFF TWO 'THREEPEAT' CHAMPIONSHIP WINS, WHILE HAVING JORDAN WIN LEAGUE MVP EACH YEAR.

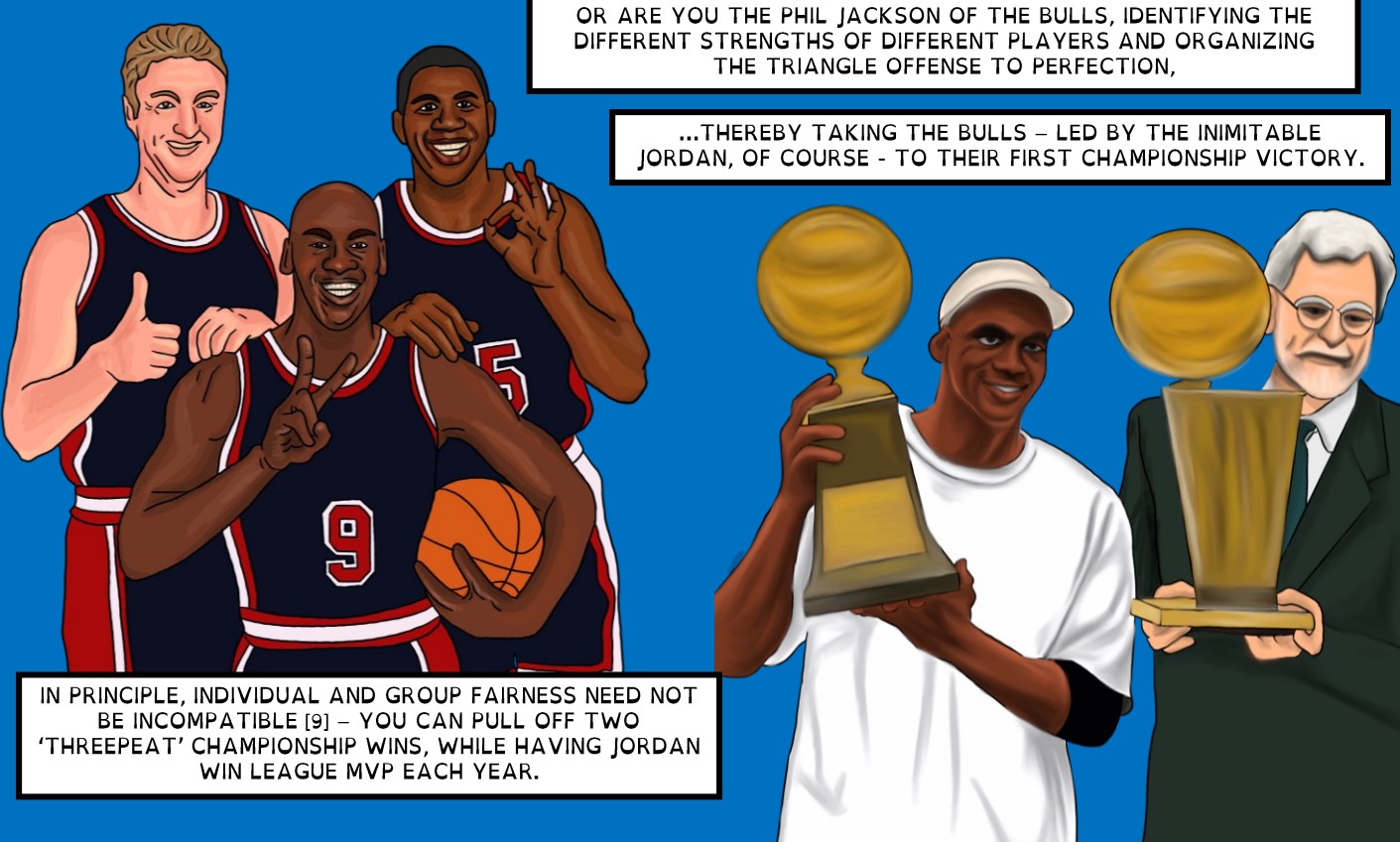

A SECOND DICHOTOMY ARISES FROM THE WAY IN WHICH WE ARRIVE AT A 'FAIR' DECISION.

**PROCEDURAL** FAIRNESS EMPHASIZES THAT THE SAME PROCESS BE APPLIED TO ALL INDIVIDUALS,

IRRESPECTIVE OF THE SOCIETAL FACTORS THAT MIGHT ADVANTAGE SOME AND DISADVANTAGE OTHERS IN GETTING A 'FAIR' SHOT IN THE SELECTION PROCESS.

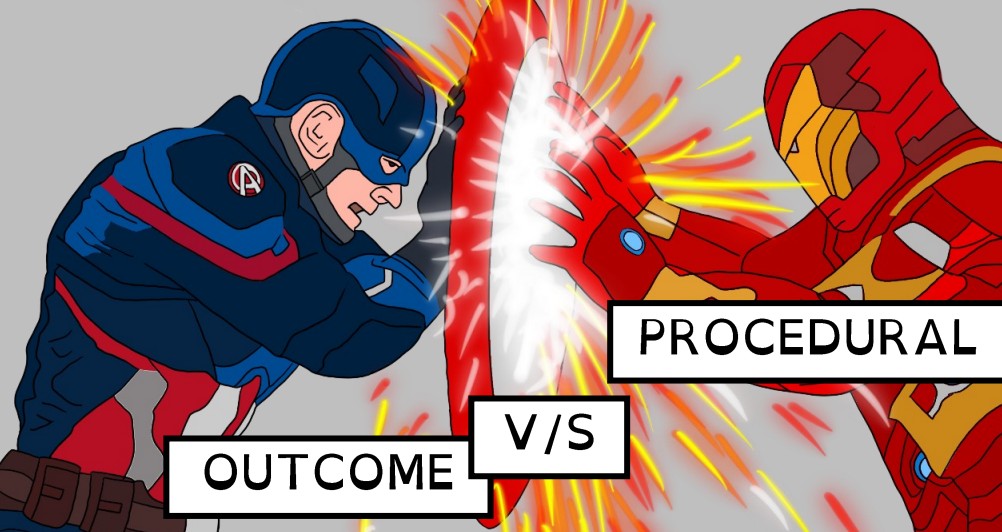

PROCEDURAL

V/S

OUTCOME

**OUTCOME** FAIRNESS, ON THE OTHER HAND, AIMS TO ENSURE THAT OUTCOMES (POSITIVE OR NEGATIVE) MEET SOME REQUIREMENT, SUCH AS POSITIVE OUTCOMES BEING DISTRIBUTED EQUALLY AMONG DIFFERENT GROUPS.

THIS ENSURES THAT MEMBERS FROM CERTAIN GROUPS ARE NOT SYSTEMATICALLY DISADVANTAGED WITH RESPECT TO OUTCOMES, BUT MIGHT COME AT THE COST OF PROCEDURAL FAIRNESS...

CORRECTING FOR SYSTEMIC INEQUALITIES MIGHT REQUIRE A DIFFERENT PROCEDURE TO BE APPLIED TO CANDIDATES FROM DIFFERENT GROUPS.

THIS DICHOTOMY TRACKS TWO DOCTRINES FROM US ANTI-DISCRIMINATION LAW - DISPARATE TREATMENT AND DISPARATE IMPACT.

**DISPARATE TREATMENT** PROHIBITS PROCEDURAL UNFAIRNESS - INTENTIONAL DISCRIMINATION THROUGH THE USE OF DIFFERENT FORMAL PROCEDURES OR MAKING DECISIONS BASED EXPLICITLY ON PROTECTED CHARACTERISTICS IS ILLEGAL.

**DISPARATE IMPACT**, ON THE OTHER HAND, PROHIBITS UNJUSTIFIED AND AVOIDABLE DISPARITIES IN OUTCOMES FOR PEOPLE OF DIFFERENT PROTECTED GROUPS.

THIS VERY DISAGREEMENT ALMOST BROKE UP THE MIGHTY AVENGERS!

ON ONE HAND, YOU HAVE TEAM STARK, WHO BELIEVE IN SIGNING THE ACCORDS AND OPERATING UNDER A PRESCRIBED MANDATE AND PROCEDURE.

AND THEN THERE ARE THOSE WHO, LIKE CAP, BELIEVE IN THE EFFICACY OF THE OUTCOME, EVEN IF IT REQUIRES PREFERENTIAL TREATMENT.

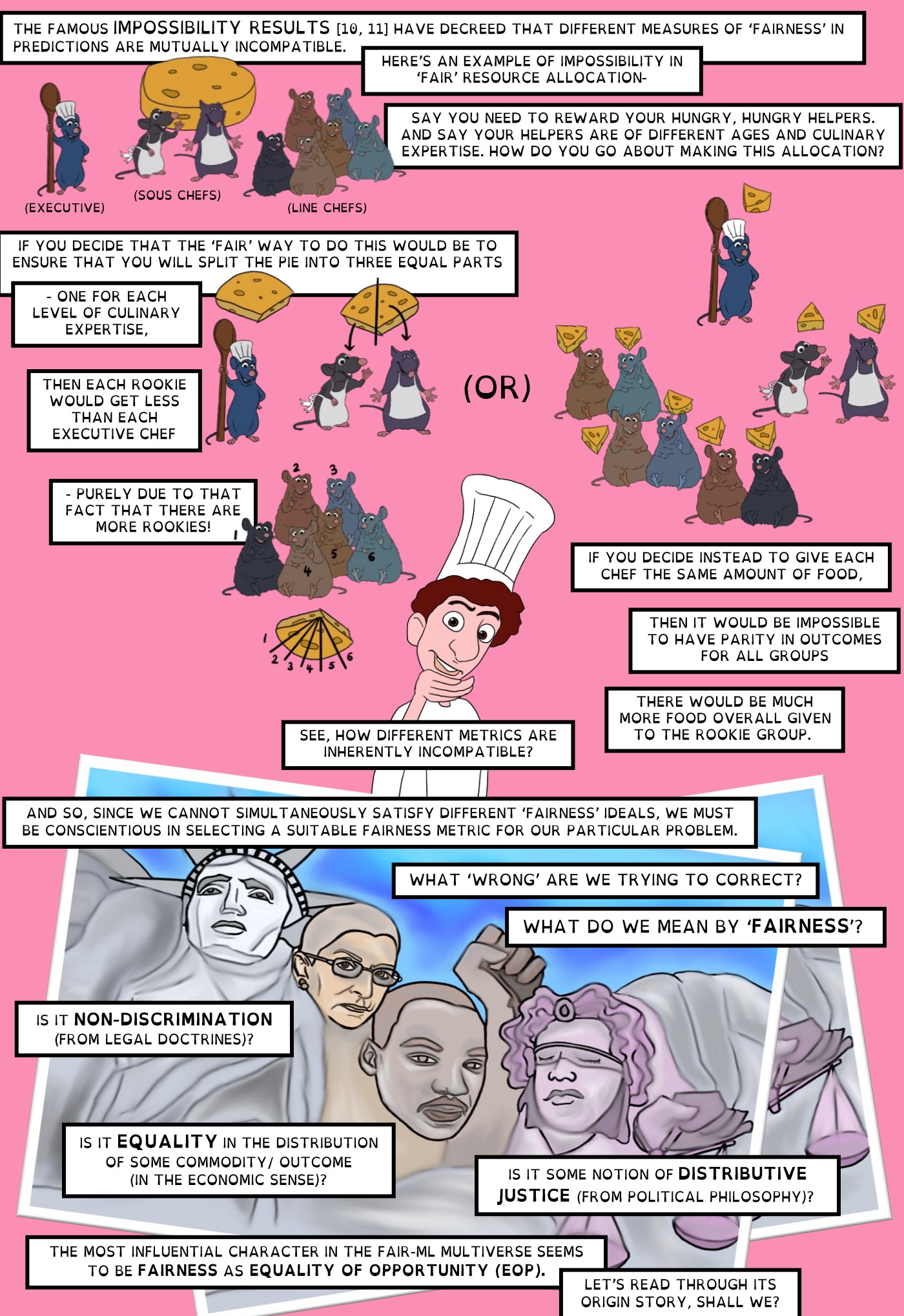

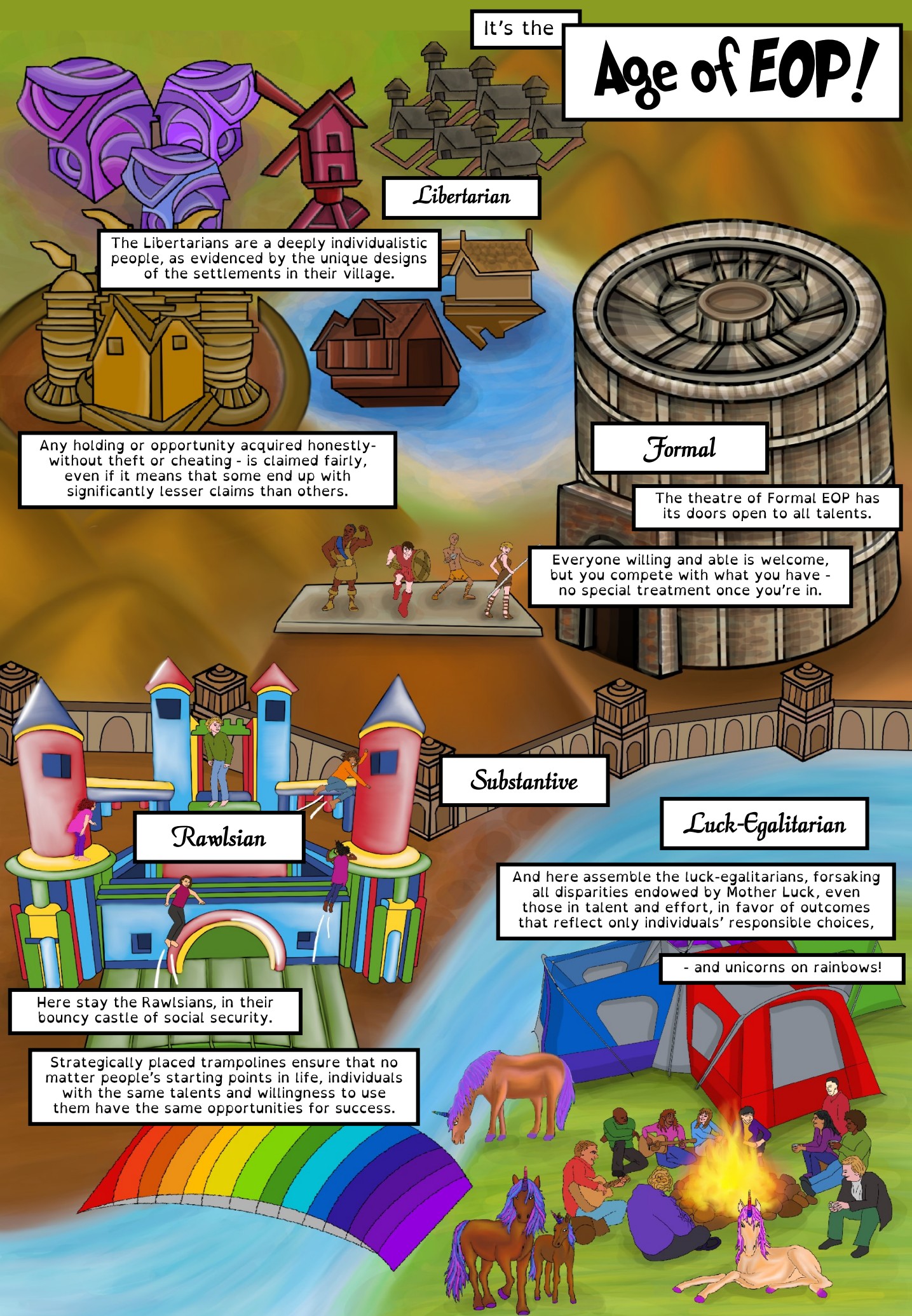

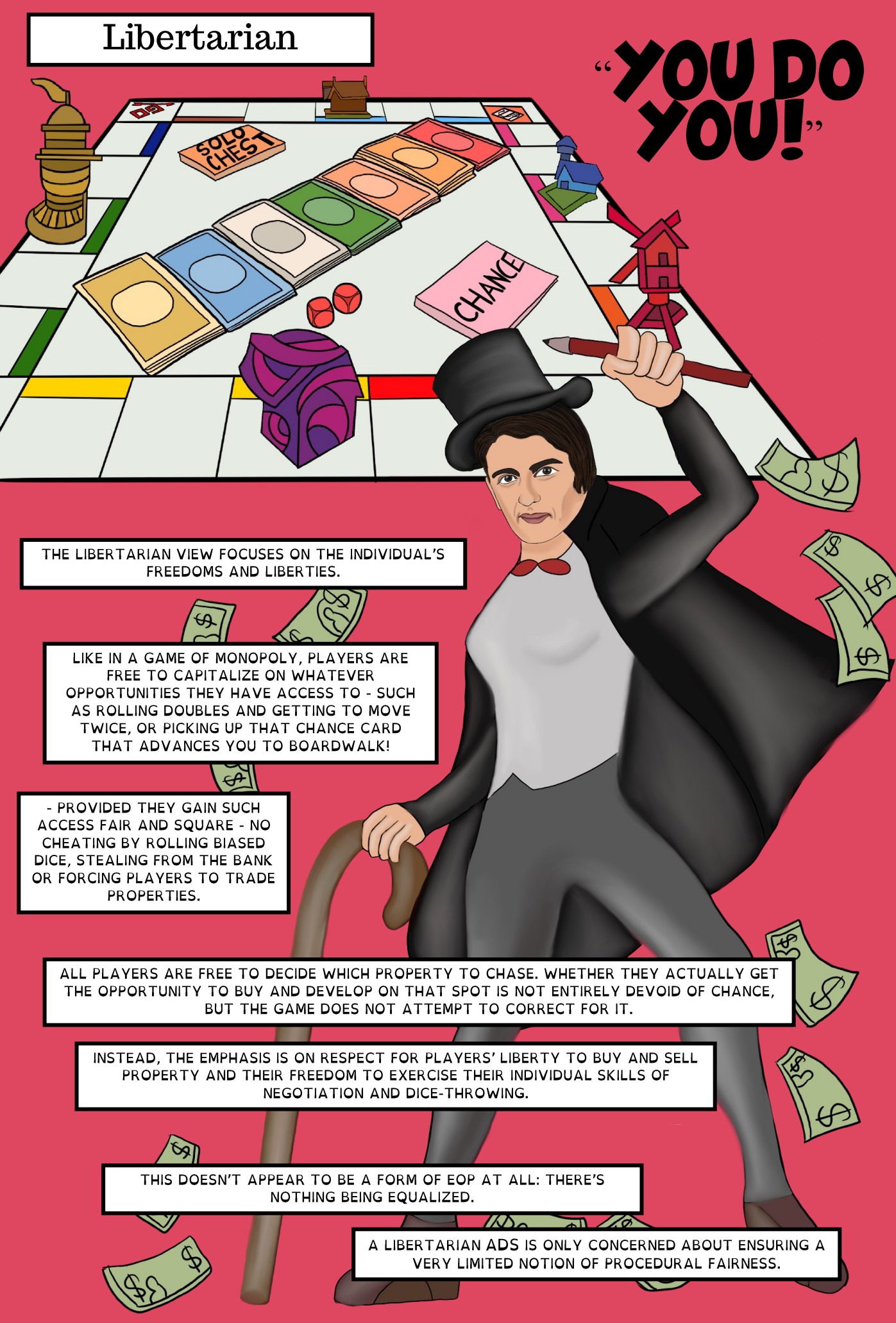

## "CAREERS OPEN TO TALENTS"

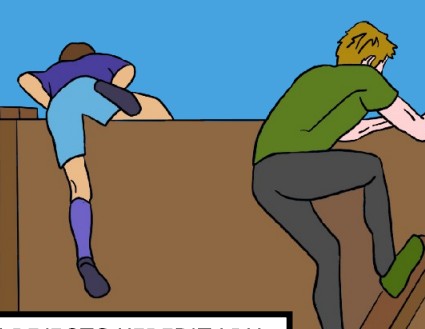

FORMAL EOP SAYS A COMPETITION IS FAIR WHEN COMPETITORS ARE ONLY EVALUATED ON THE BASIS OF THEIR RELEVANT QUALIFICATIONS - IN ANY CONTEST, THE MOST QUALIFIED PERSON WINS.

THIS IS A VIEW THAT REJECTS HEREDITARY PRIVILEGE AS THE BASIS FOR WINNING POSITIONS: BEING AN ARISTOCRAT WON'T GET YOU THE JOB.

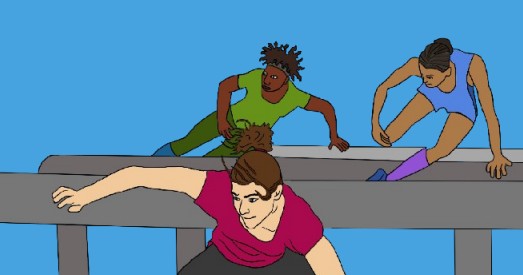

STILL, FORMAL EOP MAKES NO ATTEMPT TO CORRECT FOR ARBITRARY PRIVILEGES AND DISADVANTAGES THAT CAN LEAD TO DISPARITIES IN INDIVIDUALS' OPPORTUNITIES TO BUILD QUALIFICATIONS.

FORMAL EOP ADVOCATES 'SEE NOTHING IRRELEVANT, SPEAK NOTHING IRRELEVANT, HEAR NOTHING IRRELEVANT'.

DECISION MAKERS ARE TAUGHT TO IGNORE IRRELEVANT TRAITS LIKE SOCIAL STATUS AND TO FOCUS ONLY ON RELEVANT QUALIFICATIONS IN ADJUDICATING A CONTEST

IN FAIR-ML, THIS HAS BEEN CODIFIED AS 'FAIRNESS THROUGH BLINDNESS', WHERE ANY PROTECTED ATTRIBUTES - THOSE THAT CAN IDENTIFY GROUP MEMBERSHIP - ARE STRIPPED AWAY FROM THE DATA.

BUT THERE'S MORE TO FORMAL EOP, IF WE CONSIDER ITS MOTIVATION. A TEST THAT IS MORE INACCURATE FOR MEMBERS OF A PROTECTED CLASS - THAT BADLY MISMEASURES THE QUALIFICATIONS OF WOMEN CANDIDATES COMPARED TO MEN, FOR EXAMPLE - ALSO VIOLATES THE SPIRIT OF FORMAL EOP, EVEN IF THE TEST DOES NOT TAKE GENDER INTO ACCOUNT. [12]

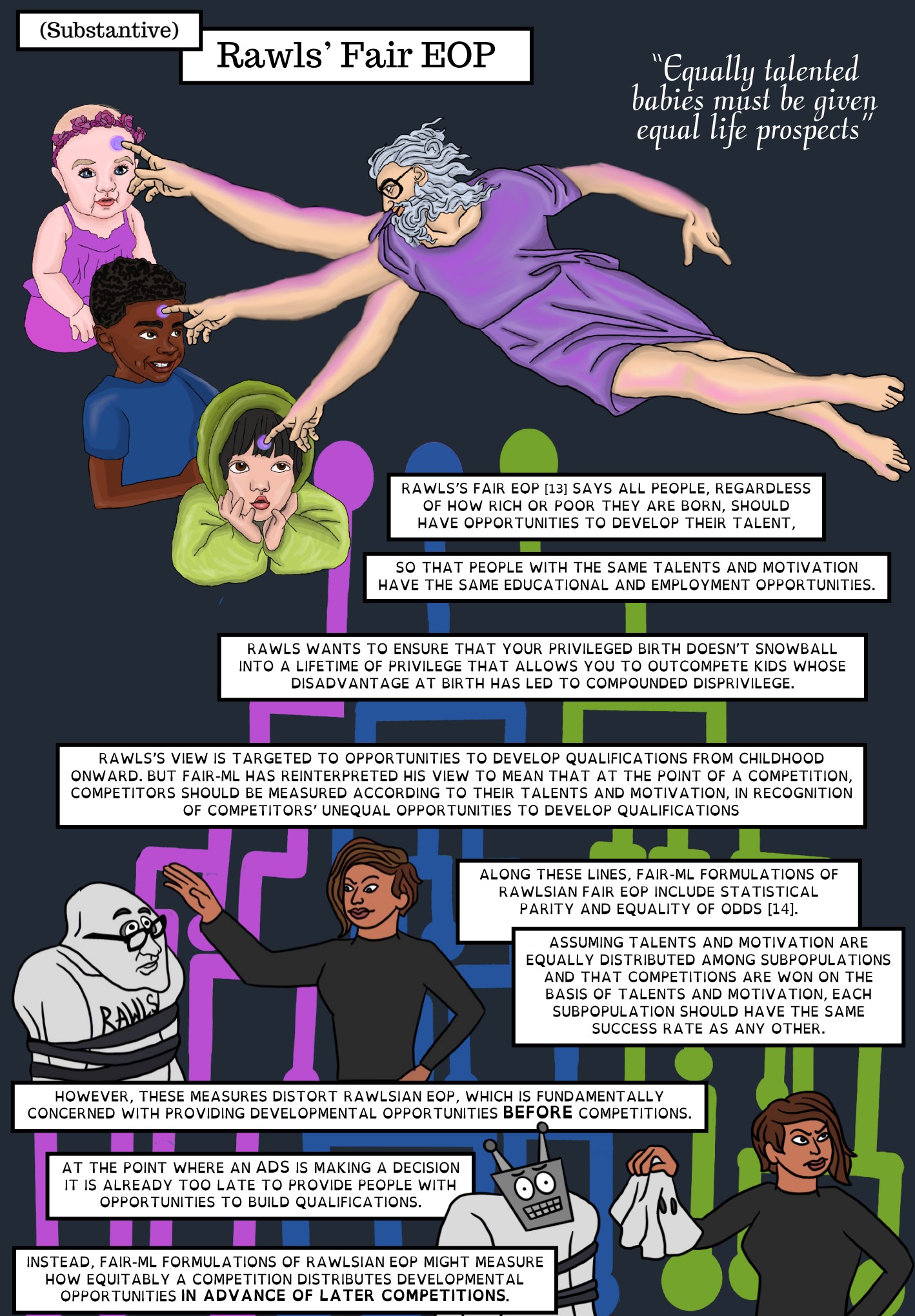

(Substantive)

# Rawls' Fair EOP

*"Equally talented babies must be given equal life prospects"*

RAWLS'S FAIR EOP [13] SAYS ALL PEOPLE, REGARDLESS OF HOW RICH OR POOR THEY ARE BORN, SHOULD HAVE OPPORTUNITIES TO DEVELOP THEIR TALENT,

SO THAT PEOPLE WITH THE SAME TALENTS AND MOTIVATION HAVE THE SAME EDUCATIONAL AND EMPLOYMENT OPPORTUNITIES.

RAWLS WANTS TO ENSURE THAT YOUR PRIVILEGED BIRTH DOESN'T SNOWBALL INTO A LIFETIME OF PRIVILEGE THAT ALLOWS YOU TO OUTCOMPETE KIDS WHOSE DISADVANTAGE AT BIRTH HAS LED TO COMPOUNDED DISPRIVILEGE.

RAWLS'S VIEW IS TARGETED TO OPPORTUNITIES TO DEVELOP QUALIFICATIONS FROM CHILDHOOD ONWARD. BUT FAIR-ML HAS REINTERPRETED HIS VIEW TO MEAN THAT AT THE POINT OF A COMPETITION, COMPETITORS SHOULD BE MEASURED ACCORDING TO THEIR TALENTS AND MOTIVATION, IN RECOGNITION OF COMPETITORS' UNEQUAL OPPORTUNITIES TO DEVELOP QUALIFICATIONS

ALONG THESE LINES, FAIR-ML FORMULATIONS OF RAWLSIAN FAIR EOP INCLUDE STATISTICAL PARITY AND EQUALITY OF ODDS [14].

ASSUMING TALENTS AND MOTIVATION ARE EQUALLY DISTRIBUTED AMONG SUBPOPULATIONS AND THAT COMPETITIONS ARE WON ON THE BASIS OF TALENTS AND MOTIVATION, EACH SUBPOPULATION SHOULD HAVE THE SAME SUCCESS RATE AS ANY OTHER.

HOWEVER, THESE MEASURES DISTORT RAWLSIAN EOP, WHICH IS FUNDAMENTALLY CONCERNED WITH PROVIDING DEVELOPMENTAL OPPORTUNITIES **BEFORE** COMPETITIONS.

AT THE POINT WHERE AN ADS IS MAKING A DECISION IT IS ALREADY TOO LATE TO PROVIDE PEOPLE WITH OPPORTUNITIES TO BUILD QUALIFICATIONS.

INSTEAD, FAIR-ML FORMULATIONS OF RAWLSIAN EOP MIGHT MEASURE HOW EQUITABLY A COMPETITION DISTRIBUTES DEVELOPMENTAL OPPORTUNITIES **IN ADVANCE OF LATER COMPETITIONS.**

# Luck-Egalitarian EOP

## "Nothing that you did not choose for yourself should affect your life prospects"

THE LUCK EGALITARIAN SAYS THAT RAWLS DOESN'T GO FAR ENOUGH IN CONTROLLING FOR FACTORS THAT PROVIDE UNFAIR ADVANTAGE OR DISADVANTAGE.

OUR OUTCOMES SHOULD ONLY BE AFFECTED BY OUR "CHOICE LUCK" (RESPONSIBLE CHOICES); NO EFFECTS OF "BRUTE LUCK" (FROM HAVING RICH PARENTS TO GETTING STRUCK BY LIGHTNING) SHOULD BE ALLOWED TO STAND.

HOW DO WE SEPARATE THE EFFECTS OF **LUCK** FROM THE EFFECTS OF **RESPONSIBLE CHOICES**?

ONE POPULAR FORMULATION IN FAIR-ML IS **ROEMER'S EOP** [15], WHICH MEASURES A PERSON'S EFFORT COMPARED TO OTHERS IN SIMILAR CIRCUMSTANCES. [16]

THIS DIALS BACK ON THE IDEA OF CONTROLLING FOR ALL BRUTE LUCK. INSTEAD, WE FOCUS ON A FEW BRUTE LUCK FACTORS, SUCH AS RACE AND SEX, THAT TRACK SIGNIFICANT UNDESERVED PRIVILEGE AND DISPRIVILEGE AND AFFECT PEOPLE'S OPPORTUNITIES TO DEVELOP QUALIFICATIONS.

WE CREATE BRACKETS BASED ON MATTERS OF BRUTE LUCK AND THEN COMPARE CANDIDATES TO OTHERS IN THEIR OWN BRACKETS.

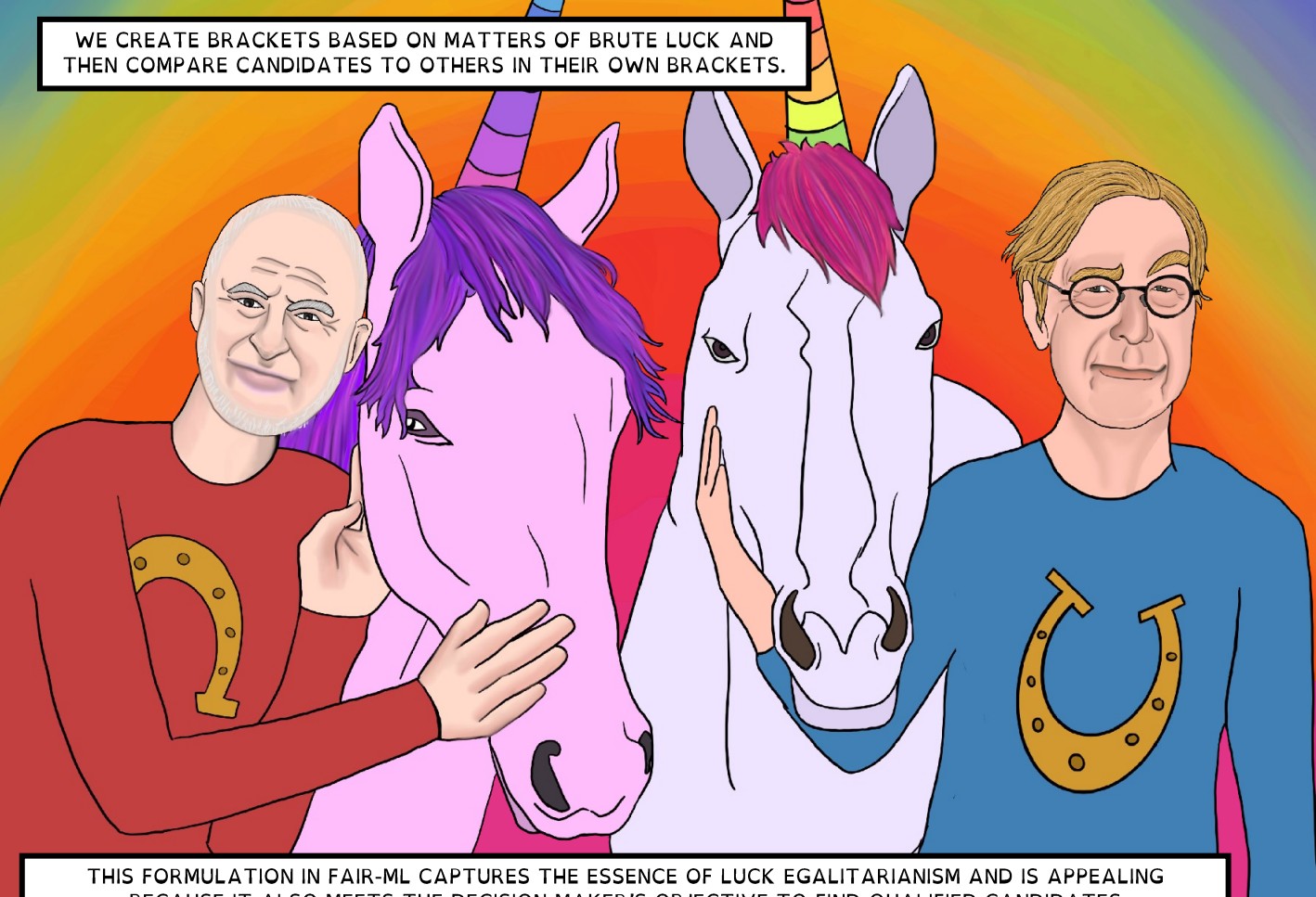

THIS FORMULATION IN FAIR-ML CAPTURES THE ESSENCE OF LUCK EGALITARIANISM AND IS APPEALING BECAUSE IT ALSO MEETS THE DECISION MAKER'S OBJECTIVE TO FIND QUALIFIED CANDIDATES

- THE ADS CONSIDERS ALL OF A CANDIDATE'S QUALIFICATIONS, NOT JUST THOSE THAT ARE ATTRIBUTABLE TO NATIVE TALENT/MOTIVATION (RAWLS) OR RESPONSIBLE CHOICES (OTHER LUCK EGALITARIANS)

THE IMPOSSIBILITY RESULTS IN FAIR-ML ARE COMMONLY INTERPRETED TO MEAN THAT 'FAIRNESS IS IMPOSSIBLE'.

BUT, IF WE LOOK AT DIFFERENT STATISTICAL MEASURES AS PROMOTING DIFFERENT CONCEPTIONS OF EOP - FORMAL VS SUBSTANTIVE, THEN THIS INCOMPATIBILITY IS WHOLLY UNSURPRISING.

WE WOULD NOT EXPECT A WORLD VIEW THAT ONLY LOOKS AT 'RELEVANT' QUALIFICATIONS AT THE POINT OF COMPETITION (FORMAL EOP) TO BE COMPATIBLE WITH ONE THAT AIMS TO PROVIDE COMPARABLE DEVELOPMENTAL OPPORTUNITIES FOR INDIVIDUALS AND, AT THE POINT OF COMPETITION, SEEKS TO CORRECT FOR INEQUALITIES IN CANDIDATES' DEVELOPMENTAL OPPORTUNITIES (SUBSTANTIVE).

WE CAN INTERPRET THIS INCOMPATIBILITY AS THE DIFFERENCE IN PHILOSOPHICAL VIEWPOINTS AND INCENTIVES OF DECISION MAKERS.

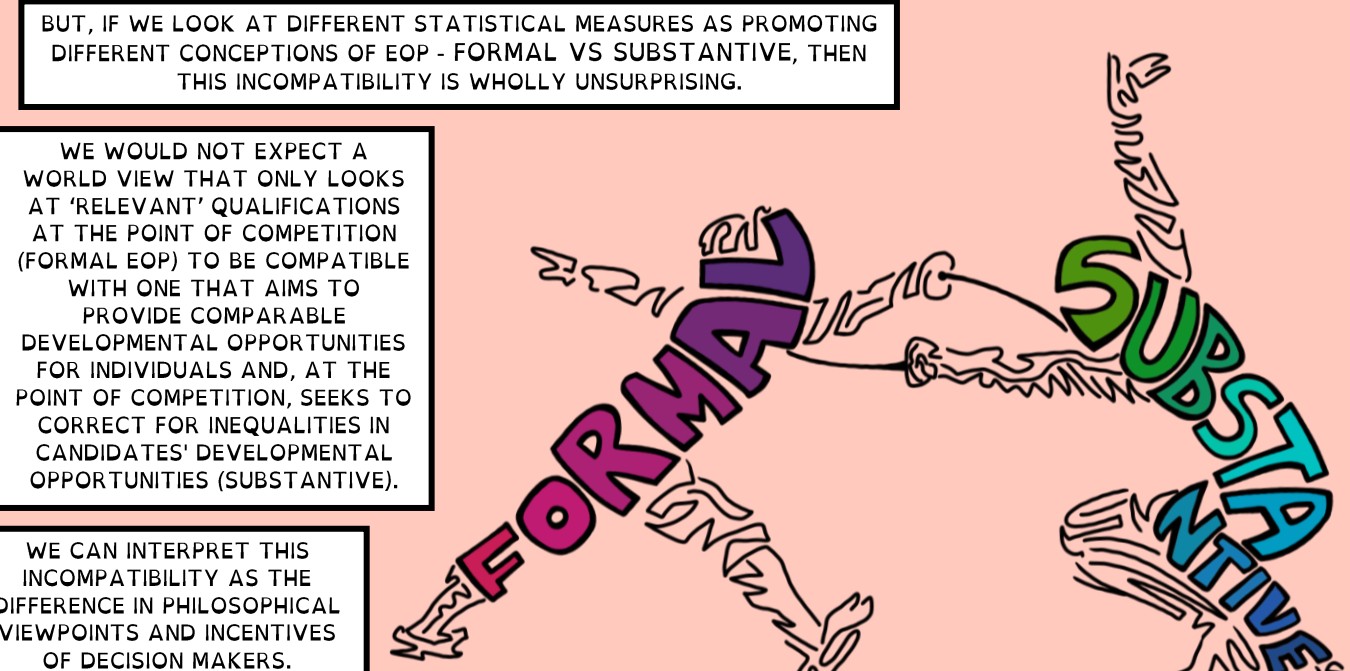

THIS GROUNDING GIVES US SOME MUCH-NEEDED GUIDANCE IN CHOOSING A SUITABLE 'FAIRNESS' MEASURE FOR OUR GIVEN CONTEXT.

IF WE BELIEVE THAT INEQUALITIES OF BIRTH DO NOT AFFECT A PERSON'S QUALIFICATIONS, THEN THE FORMAL APPROACH MIGHT BE SUFFICIENT TO MODEL A 'FAIR' FOOTRACE BETWEEN A KING AND A PEASANT.

FORMAL EOP ALSO OFFERS FAIRNESS IN THE FORM OF 'BLIND AUDITIONS'.

WHEN WE WORRY THAT JUDGES WILL BE SWAYED BY IRRELEVANT TRAITS LIKE GENDER, RACE AND APPEARANCE, BLIND AUDITIONS FORCE JUDGES TO EVALUATE CONTESTANTS SOLELY ON THEIR SINGING CHOPS.

SIMILARLY, MAKING EMPLOYERS BLIND TO JOB APPLICANTS' CREDIT SCORES OR CRIMINAL CONVICTIONS DURING INITIAL APPLICANT SCREENINGS CAN HELP PEOPLE OVERCOME STUBBORN OBSTACLES TO EMPLOYMENT!

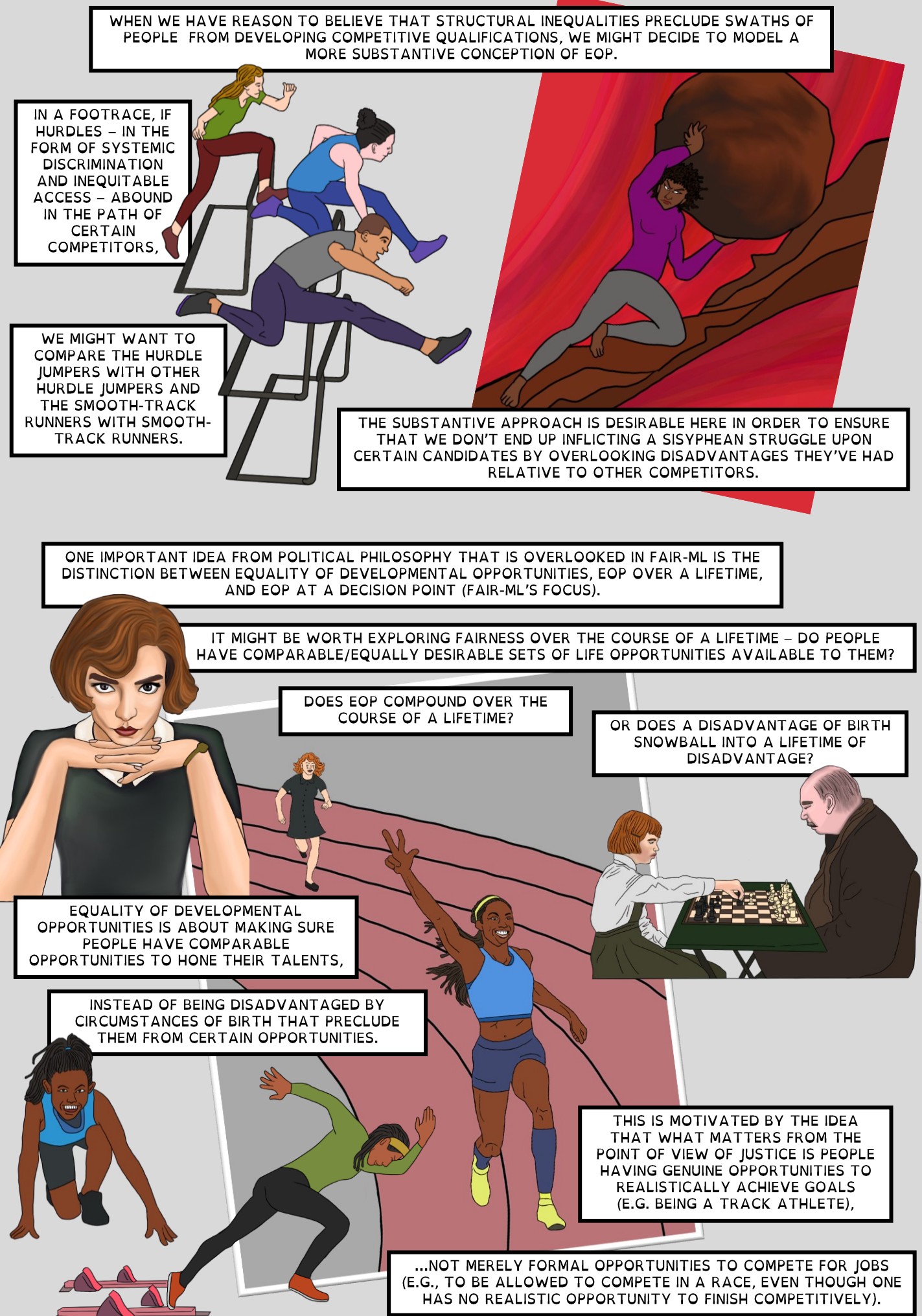

OUR STROLL THROUGH EOP-VILLE HAS SHOWN US A RANGE OF INTERPRETATIONS OF 'FAIRNESS'. BUT IS 'FAIRNESS' ALL THAT'S REQUIRED FOR AN ALGORITHM TO BE 'JUST'?

RAWLS SANDWICHES HIS EOP PRINCIPLE BETWEEN TWO OTHER PRINCIPLES THAT ALSO MUST BE SATISFIED FOR A DEMOCRATIC SOCIETY TO BE 'JUST'.

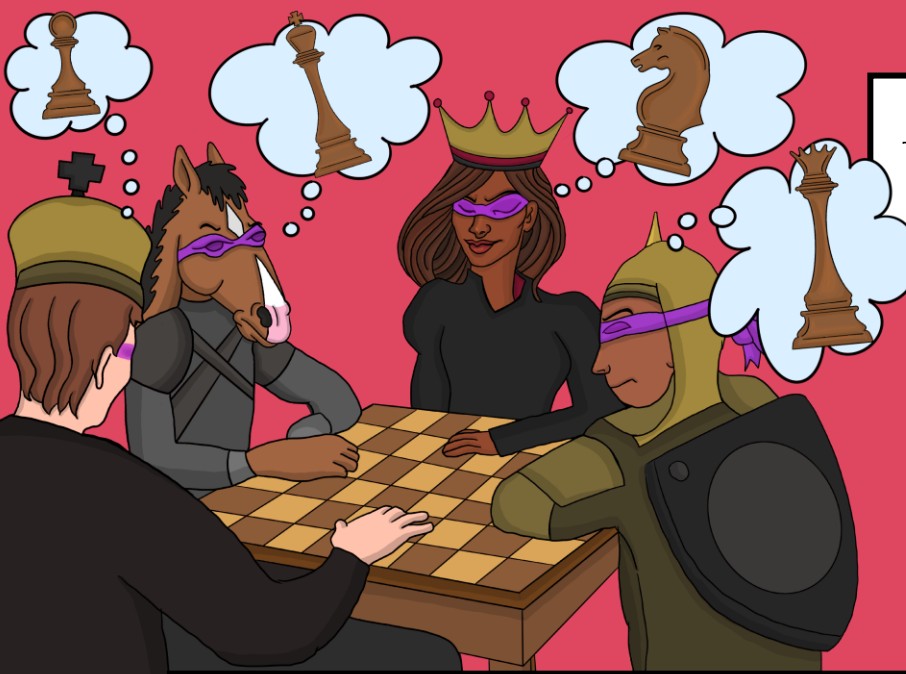

HE ARRIVES AT THESE PRINCIPLES VIA **THE ORIGINAL POSITION**- A THOUGHT EXPERIMENT ABOUT HOW CITIZENS WOULD NEGOTIATE THE SET-UP OF SOCIETY, UNDER THE **'VEIL OF IGNORANCE'**

- IF CITIZENS DO NOT KNOW THEIR RACE, CLASS, SEX, TALENTS, SOCIAL POSITION (OR ANY OTHER CHARACTERISTICS THAT MIGHT CAUSE THEM TO FAVOR PEOPLE LIKE THEMSELVES), THEY WILL ADVOCATE FOR ALL SOCIAL POSITIONS AND THEIR ATTACHED PRIVILEGES TO BE DISTRIBUTED 'FAIRLY'.

BUT THEY DO KNOW THAT PEOPLE ARE FREE AND EQUAL AND THAT THEY HAVE THE ABILITY TO CHOOSE A CONCEPTION OF THE GOOD LIFE AND THE ABILITY TO ABIDE BY RULES OF JUSTICE.

AND SO, RAWLS POSITS THAT THE PRINCIPLES OF SOCIAL COOPERATION THAT PEOPLE ARRIVE AT THROUGH SUCH A NEGOTIATION WILL BE APPROPRIATE FOR A FREE AND DEMOCRATIC SOCIETY.

RAWLS USES THE NOTION OF THE **"NATURAL LOTTERY"** TO DESCRIBE THE MORALLY ARBITRARY DISTRIBUTION OF TALENTS, FAMILY CIRCUMSTANCES, AND OTHER AT-BIRTH FORTUNE AND MISFORTUNE TO PEOPLE.

FROM THE ARBITRARINESS OF THE NATURAL LOTTERY, RAWLS CONCLUDES THAT WE DON'T DESERVE OUR STARTING POINTS IN LIFE,

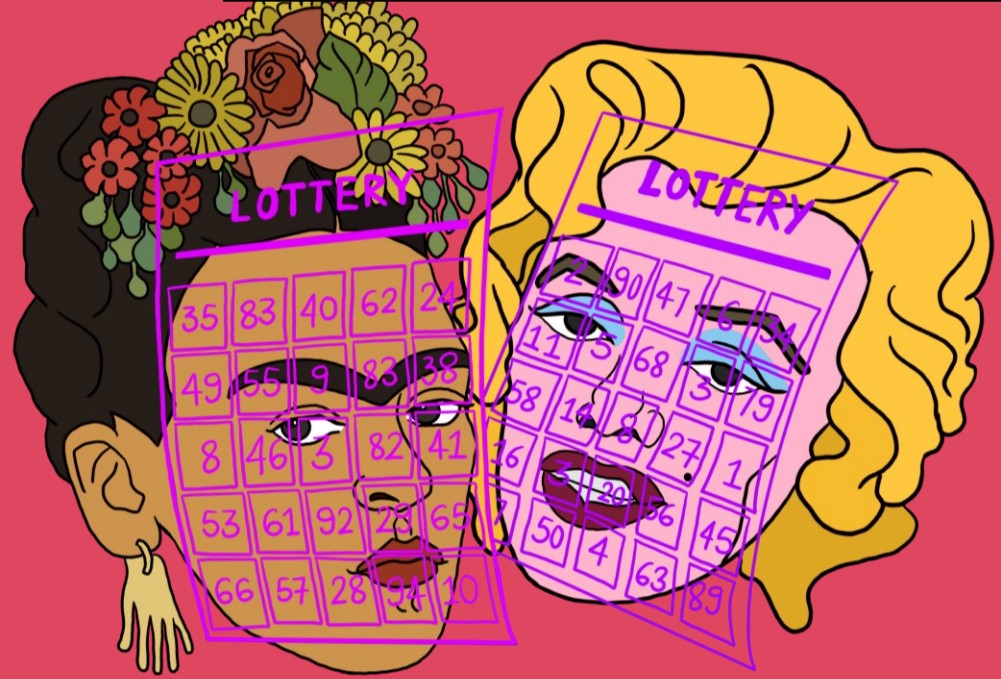

...AND ARRIVES AT THE **DIFFERENCE PRINCIPLE** - WHICH HARNESSES THE ARBITRARY DISTRIBUTION OF TALENTS TO GENERATE A SOCIAL SYSTEM THAT SERVES EVERYONE.

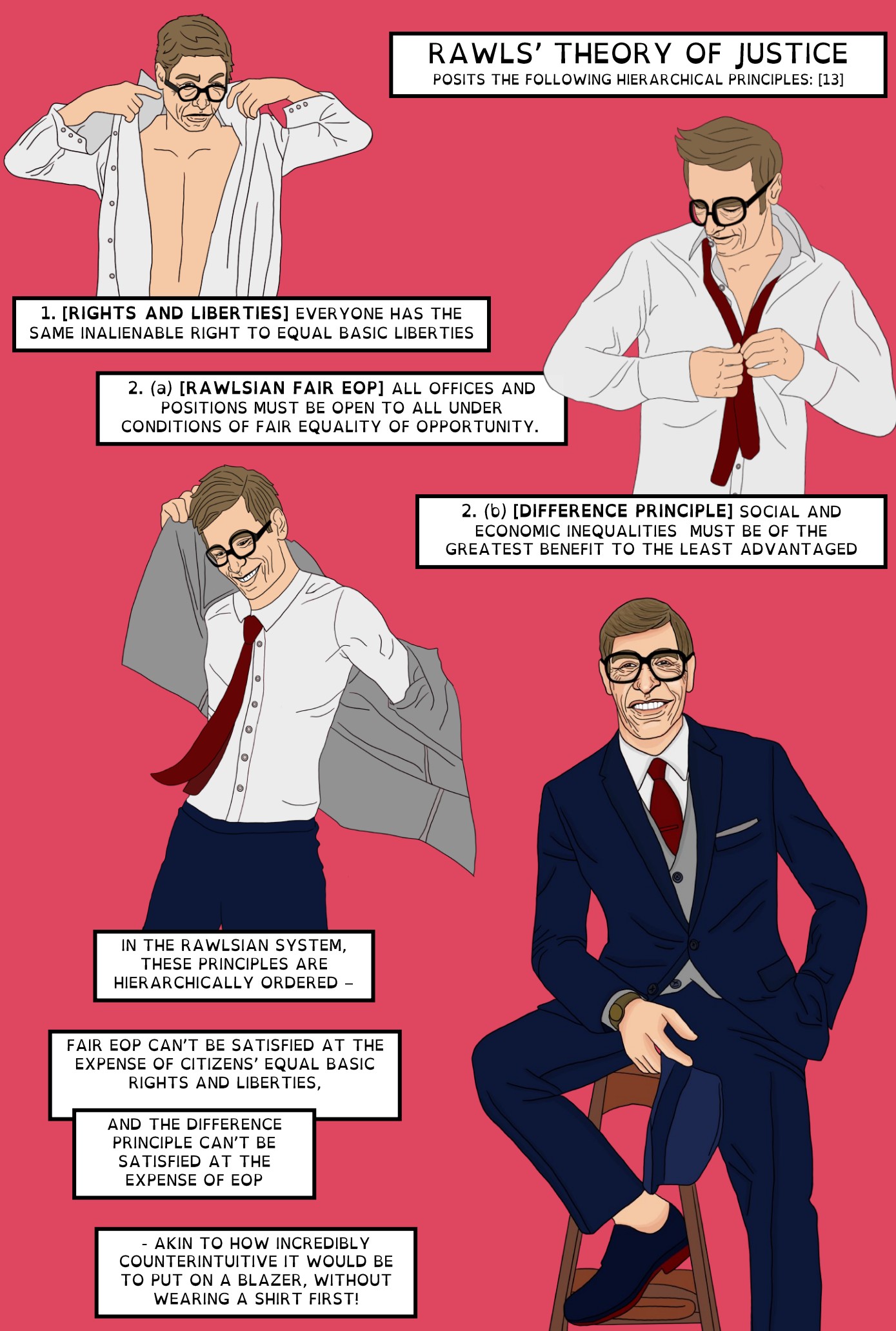

# RAWLS' THEORY OF JUSTICE
POSITS THE FOLLOWING HIERARCHICAL PRINCIPLES: [13]

1. [RIGHTS AND LIBERTIES] EVERYONE HAS THE SAME INALIENABLE RIGHT TO EQUAL BASIC LIBERTIES

2. (a) [RAWLSIAN FAIR EOP] ALL OFFICES AND POSITIONS MUST BE OPEN TO ALL UNDER CONDITIONS OF FAIR EQUALITY OF OPPORTUNITY.

2. (b) [DIFFERENCE PRINCIPLE] SOCIAL AND ECONOMIC INEQUALITIES MUST BE OF THE GREATEST BENEFIT TO THE LEAST ADVANTAGED

IN THE RAWLSIAN SYSTEM, THESE PRINCIPLES ARE HIERARCHICALLY ORDERED –

FAIR EOP CAN'T BE SATISFIED AT THE EXPENSE OF CITIZENS' EQUAL BASIC RIGHTS AND LIBERTIES,

AND THE DIFFERENCE PRINCIPLE CAN'T BE SATISFIED AT THE EXPENSE OF EOP

- AKIN TO HOW INCREDIBLY COUNTERINTUITIVE IT WOULD BE TO PUT ON A BLAZER, WITHOUT WEARING A SHIRT FIRST!

FOR EXAMPLE, TAKE THE CHILDREN OF RICH PARENTS -

IN TRYING TO GIVE PEOPLE ACCESS TO EQUAL DEVELOPMENTAL OPPORTUNITIES, ONE MIGHT END UP PREVENTING PARENTS FROM RAISING KIDS ACCORDING TO THEIR VALUES,

BECAUSE THIS WOULD MEAN THAT SOME KIDS GET BETTER DEVELOPMENTAL OPPORTUNITIES THAT OTHERS.

IN TRYING TO SATISFY RAWLS'S FAIR EOP, WE MIGHT END UP INFRINGING ON RICH PARENTS' BASIC LIBERTIES.

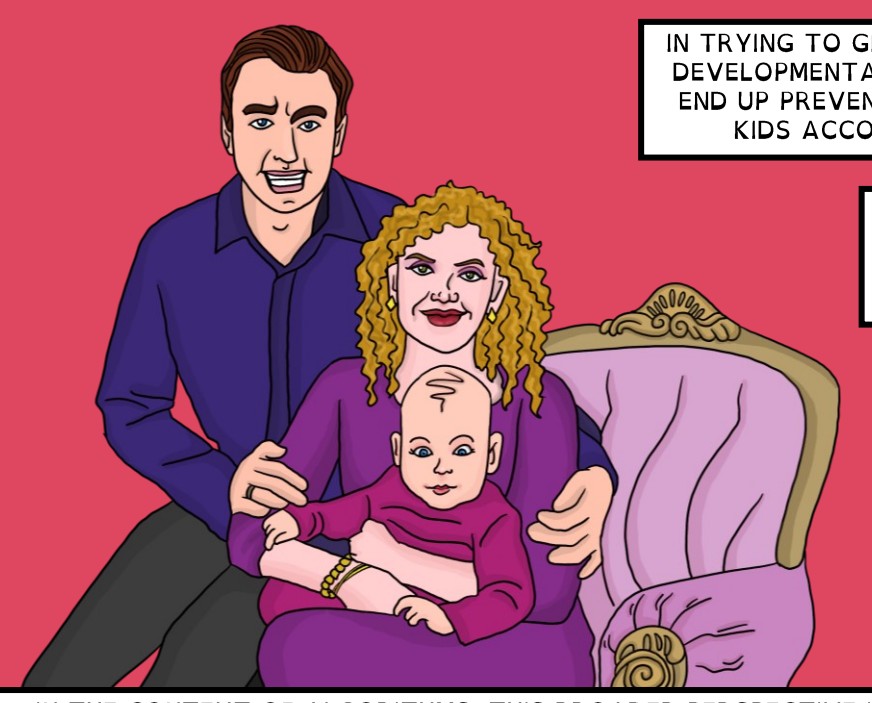

IN THE CONTEXT OF ALGORITHMS, THIS BROADER PERSPECTIVE IS HELPFUL TO SEE HOW AN ADS THAT IS (STATISTICALLY) 'FAIR' CAN GO ON TO INFRINGE ON BASIC RIGHTS AND LIBERTIES AND, IN EFFECT, BE UNJUST.

TAKE THE EXAMPLE OF "FAIR" HIRING OF PEOPLE WITH **DISABILITIES.**

"DISABILITY" WOULD BE TREATED AS A PROTECTED CLASS AND REMOVED FROM EXPLICIT CONSIDERATION,

BUT ALGORITHMS COULD STILL INFER DISABILITY FROM OTHER PROXY VARIABLES.

IF SOCIAL MEDIA INFORMATION IS USED, THE ADS COULD INFER DISABILITY STATUS—

FOR EXAMPLE, BASED ON MEMBERSHIP IN CERTAIN SOCIAL GROUPS OR ON POSTING ABOUT DISABILITY-RELATED ISSUES—

THEN A SCHEME THAT DISCRIMINATES ON THE BASIS OF "INFERRED" DISABILITY WOULD INCENTIVIZE PEOPLE AGAINST JOINING SUCH GROUPS AND SPEAKING ABOUT SUCH TOPICS.

SUCH AN ADS COULD SATISFY SOME CONCEPTION OF 'FAIRNESS' AS EOP AND YET BE FUNDAMENTALLY UNJUST: IT WOULD VIOLATE A CANDIDATE'S FREEDOM OF SPEECH AND FREEDOM OF ASSOCIATION.

THERE ARE LIMITATIONS TO WHAT ANSWERS WE CAN GET FROM EOP DOCTRINES,

AND OVERLOOKING THESE CAN EMBOLDEN THEIR APPLICATION IN SPHERES IN WHICH THEORY PROVIDES LITTLE TO NO GUIDANCE...

THESE DOCTRINES DO NOT GIVE US ANY DIRECTION ABOUT *WHERE* TO APPLY 'FAIRNESS' - IN THE PROCEDURE OR AT THE OUTCOME.

THE GUIDANCE IS ONLY ABOUT *HOW* A 'FAIR' TEST SHOULD BEHAVE.

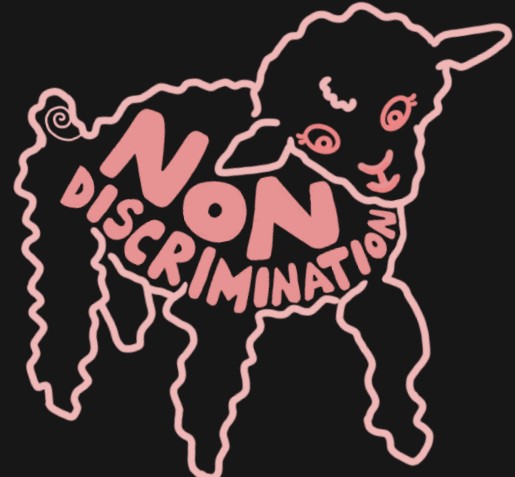

WHEN APPLYING THIS TEST TO BLACK BOX ADS, WE RUN INTO ISSUES OF INTERPRETABILITY

AND CAN ONLY INFER DETAILS ABOUT HOW THE TEST IS BEHAVING BY LOOKING AT WHICH INPUTS HAVE BEEN FED INTO THE ALGORITHM,

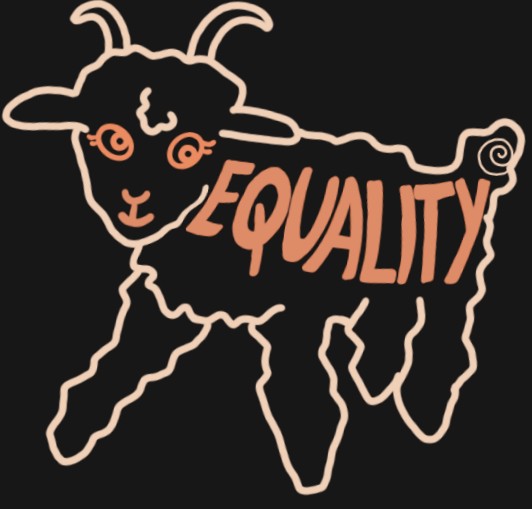

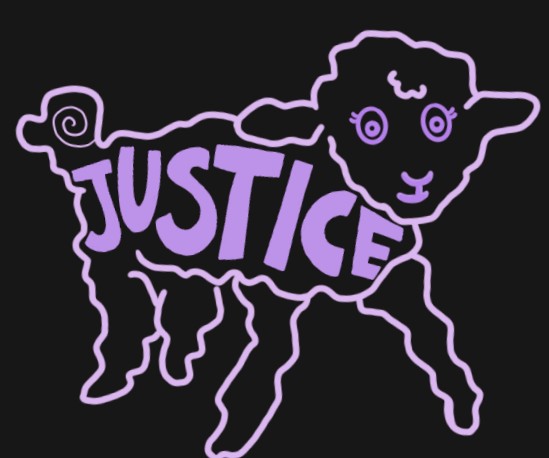

OR BY SYSTEMATICALLY STUDYING THE OUTCOMES FOR A VARIETY OF CANDIDATES.

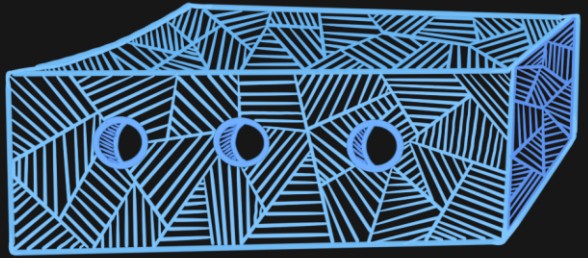

THE FAIRNESS YOU ASKED FOR IS INSIDE THIS BOX!

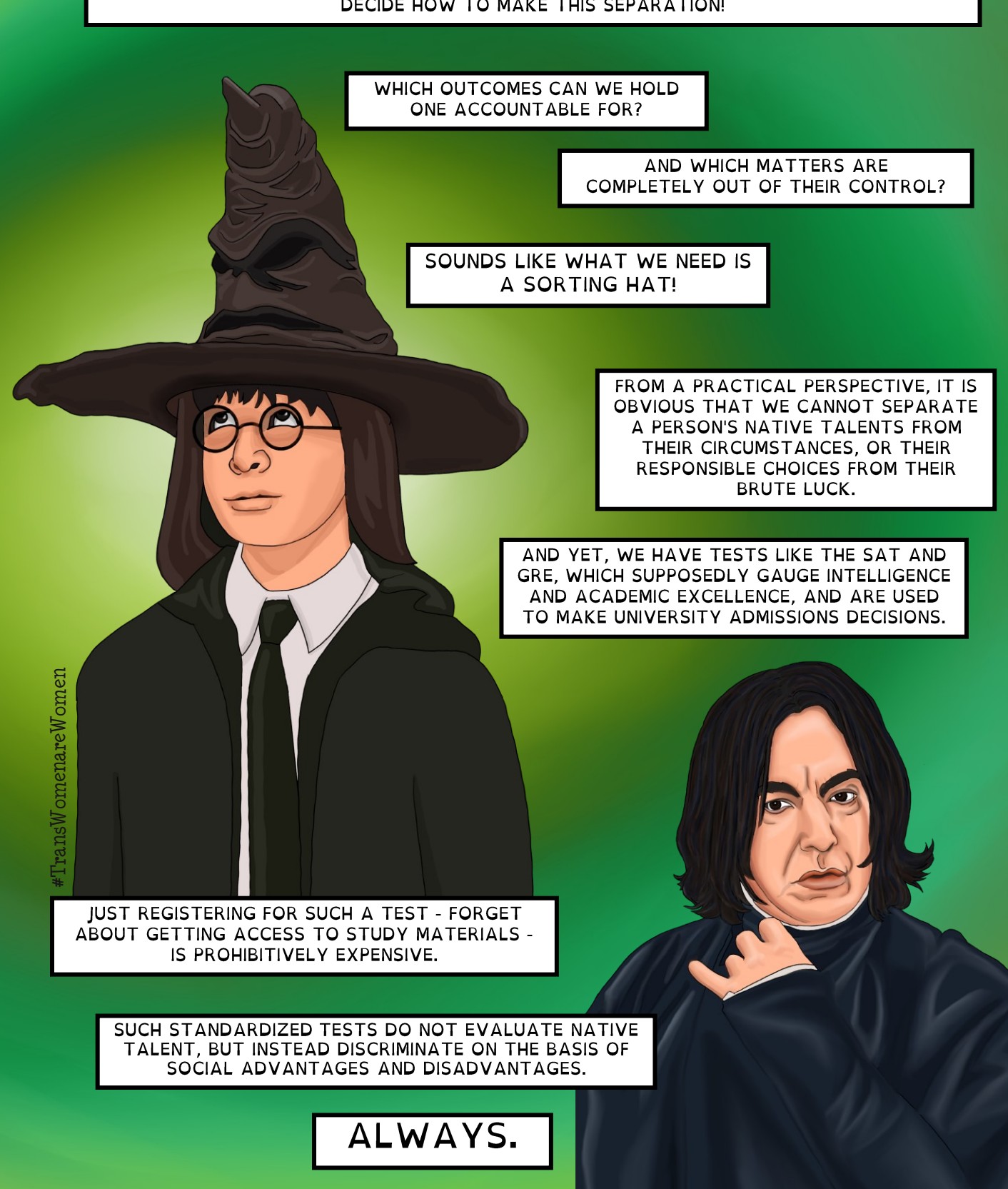

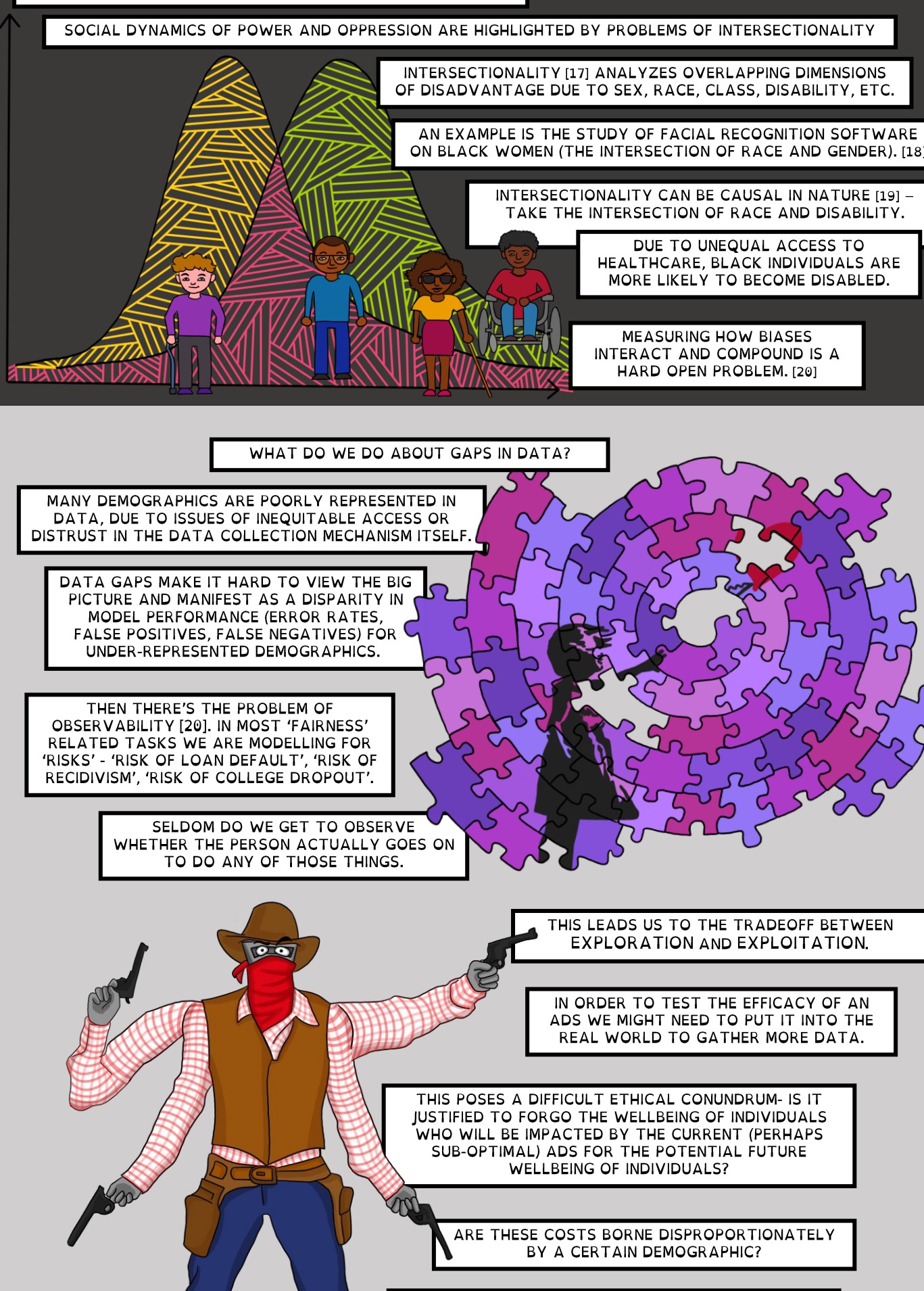

BEFORE WE DEPART, LET US HEED AN IMPORTANT WARNING ABOUT THE NATURE OF THIS TALE...

BIAS IS A THREE-HEADED DRAGON, EACH HEAD A FORMIDABLE OPPONENT IN ITS OWN RIGHT. IT'S INCREDIBLY DIFFICULT TO DETECT BIAS IN DATA, EVEN MORE SO IN THE OUTPUT OF A BLACK-BOX ML ALGORITHM.

OR WHEN THAT MODEL IS ASKED TO MAKE PREDICTIONS ON DATA THAT IS DIFFERENT FROM WHAT IT WAS TRAINED ON, POSSIBLY EVEN AS A SIDE-EFFECT OF THAT VERY MODEL'S USE.

TRAIN

THIS COMPLEXITY COMPOUNDS WHEN YOU THINK ABOUT THE INCENTIVES THAT ADS CREATE.

IT'S NOT JUST SOME ABSTRACT PREDICTION COMING OUT OF AN ALGORITHM ANYMORE

TEST

- IT'S BEING USED TO MAKE A DECISION IN THE REAL WORLD. AND THESE DECISIONS DETERMINE CRITICAL SOCIAL ALLOCATIONS SUCH AS JOBS, GRADES AND LOANS.

THIS CREATES INCENTIVES FOR PEOPLE TO BEHAVE IN A WAY THAT MAXIMIZES THEIR ALLOCATION FROM THE ADS. THIS 'NEW' BEHAVIOR IN TURN REFLECTS IN THE DATA AND AFFECTS THE SUBSEQUENT PREDICTION FROM THE ALGORITHM.

PLAYING IN THE ARENA OF FAIR-ML IS NOT ONLY LIKE FACING A THREE-HEADED DRAGON, BUT THEN HAVING A NEW, EVER-EVOLVING, DYNAMICALLY-GENERATED OPPONENT EACH TIME.

DEVISE A METHOD TO CUT OFF ONE HEAD OF PRE-EXISTING BIAS, AND TWO NEW HEADS OF EMERGENT BIASES GROW OUT.

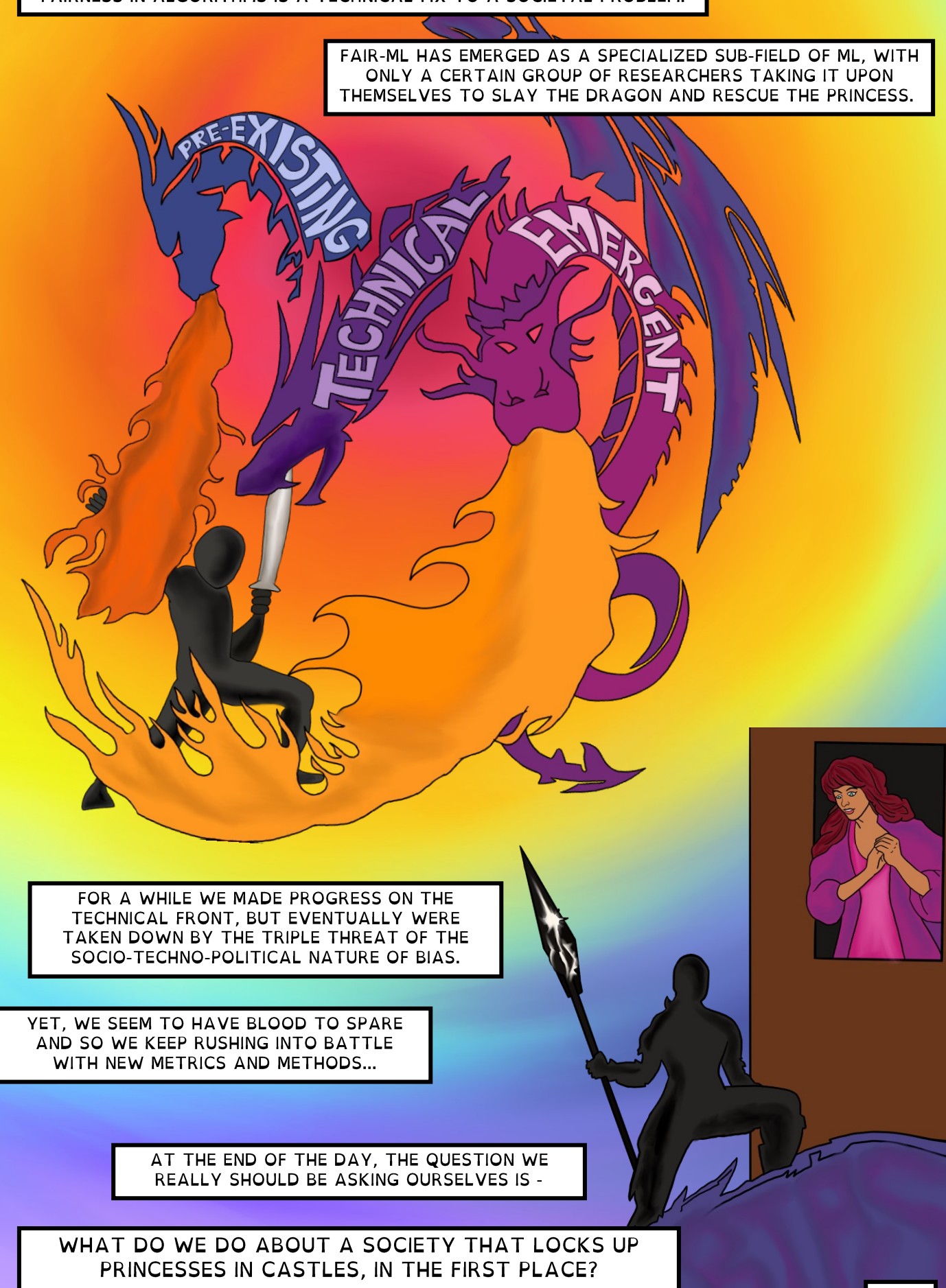

# ABOUT

A computer scientist, artist and philosopher join a zoom room. This happens!
'Fairness and Friends' is the second volume of the Data, Responsibly Comic series. We hope that it will serve as the computer scientist's guide to political philosophy!

**Falaah** is a scientist/engineer by training and an artist by nature, and the creator of MachineLearnist Comics - a collection of webcomics about the current AI landscape.

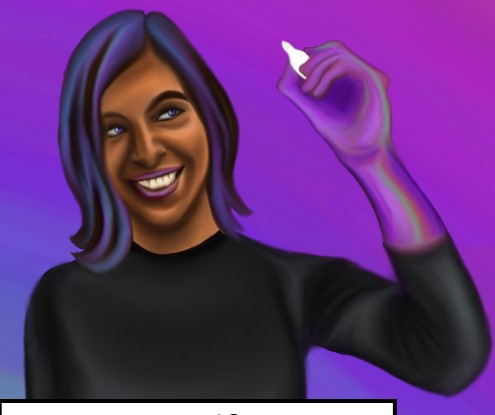

Falaah Arif Khan,
Co-Creator, Author, Artist

**Eleni** is the Research Director at the Surveillance Technology Oversight Project. She began her career as an Assistant Professor of Philosophy at Franklin & Marshall College, focused on justice in democracies, and now works at the intersection of her expertise in ethics, democratic justice, and technology policy.

Eleni Manis,
Author

**Julia** is an Assistant Professor of Computer Science and Engineering and of Data Science and the founding Director of the Center for Responsible AI at New York University. She leads the 'Data, Responsibly' project, the latest offering of which is the inimitable interdisciplinary course on Responsible Data Science.

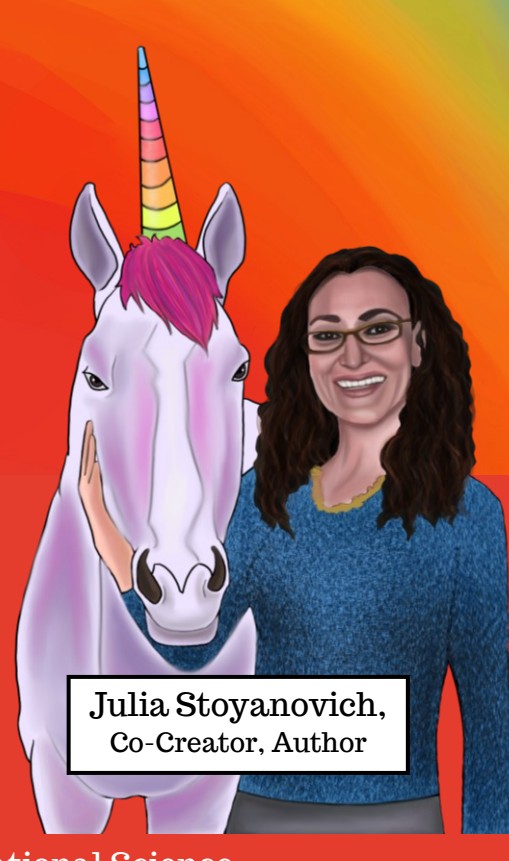

Julia Stoyanovich,
Co-Creator, Author

This work was supported in part by National Science Foundation (NSF) Grants No. 1926250, 1934464, and 1916505.

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
