# OpenReview forum: "Fairness and Friends"
_ICLR.cc/2021/Workshop/Rethinking_ML_Papers — Rethinking ML Papers - ICLR 2021 workshop Poster_

### Official Review · AnonReviewer1 · 2021-03-28
**Inspiring but overly-lengthy work**

**Accessibility:**

Score of 3 (Neutral): Submission proposes methods to improve accessibility, but the level of intended accessibility is not well-articulated. Also, the limitations and exceptions are not stated.

**Groundsforrejection:**

- Several illustrations (esp. on page 4) have religious connotations and might be a bit offensive to certain readers. Consider a more neutral tone.
- Did the authors obtain permission to depict caricatures of identifiable persons? E.g. Yann LeCun, Sundar Pichai? Otherwise, is it safe to assume press-like free-speech reproduction here?

**Litreview:**

Score of 2 (Needs Improvement): The submission leaves out prominent examples of previous work in the area.

**Problemstatement:**

Score of 4 (Strong): The submission sets a very strong example of how to address the problem, which should be relevant to the workshop themes.

**Relevance:**

Score of 5 (Exceptional): Like (4) but does so with multiple themes of the workshop.

**Results:**

Score of 3 (Neutral): Submission is well designed and provides a good level of coherency/novelty/interactivity.

**Reviewerconfidence:**

4 - Confident.
The authors draw attention to comics as a plausible medium to communicate ML findings - a great fit fo the workshop. It can inspire many papers to have a few comics that provide the big picture or communicate important points.


**Reviewtext:**

The authors use bias and fairness to demonstrate the use of comics as a new format to disseminate ML research.
The comics nicely illustrate different forms of bias as well as different schools of defining fairness and related aspects.

The comics are engaging and visually pleasing.
My major concerns is that the exhibit lacks a structure despite being very long (31+ 4 pages). I needed to go up and down several times to recover the line of thought.  Part of the reason is the depth at which the authors wanted to cover the subject.
This does not mean that comics inherently lack structure. For example, light-weight comics-like illustration have been nicely embedded in this article on BERT compression: https://blog.rasa.com/compressing-bert-for-faster-prediction-2/
Also, 1-page XKCD comics often effectively communicate important subtitles and nuances in ML:
https://xkcd.com/1725/
https://xkcd.com/2048/
https://xkcd.com/605/

My other concern is the lack of related work on using comics in ML, as in the above XKCD and blog posts.
Needless to say, other branches of science have long tradition in using comics - I do not expect the authors to reference all such pieces, but it could be useful to frame their work as part of a wider effort and initiative, e.g. the extensive work by Jorge Cham:
- Gravitational Waves: http://phdcomics.com/comics.php?f=1853
- Higgs Boson: http://phdcomics.com/comics.php?f=1489
- We Have No Idea (book): http://phdcomics.com/noidea/?ref=PRHC447FFF6CAF2
Finally, the collection at IDEA Instructions (https://idea-instructions.com/) illustrates the use of IKEA-style (assembly) instructions to communicate how algorithms (and other Computer Science concepts) work.

Finally, it could be helpful to have a high-level discussion on the implications of comics as a medium in ML research. E.g. they can be useful when comparing competing approaches, architectures, losses, regularization schemes, etc. as the authors demonstrated nicely with fairness definitions.

**Score:**

Accept: The reviewer believes the submission provides a novel and reliable scheme to improve science communication but needs improvement.

---

### Official Review · AnonReviewer2 · 2021-03-28
**Neat work!**

**Accessibility:**

Score of 4 (Strong): Submission states accessibility concerns and provides solutions within the proposed framework. However, it does not declare the limitations and exceptions.

**Groundsforrejection:**

N/A

**Litreview:**

Score of 4 (Strong): The submission directly differentiates itself from previous works and formats.

**Problemstatement:**

Score of 4 (Strong): The submission sets a very strong example of how to address the problem, which should be relevant to the workshop themes.

**Relevance:**

Score of 4 (Strong): The submission directly addresses a theme of the workshop, and does so in a very professional manner.

**Results:**

Score of 4 (Strong): Submission is very well structured and follows all the criteria (i.e. clarity, novelty, interactivity, and coherency). However, practical significance/theoretical implications are not discussed.

**Reviewerconfidence:**

- I am a voracious consumer of comics and zines
- I am, well, in some ways involved with AI-ethics

Hence, clearly, my confidence is my provided judgement is 2.71828182845904523536028747135266249775724709369995




**Reviewtext:**

It was heartening to see this beautiful piece of work. Kudos!
 I particularly enjoyed the iconoclastic take against the institution of Godfathership, which, IMHO is hurting our community & encouraging acerbic narratives.
1: Color choice: I opened the pdf on a 27" CF398 monitor & the purple hued pages such as  2, 3, 11 & 32 were rather challenging to view. I suppose the effect would be different on printed pages. Given that I saw digital artists amongst the authors' list, I am confident there was much thought put into the color choices, & the viewing discomfort could entirely be on account of my idiosyncratic local-monitor settings. I'd be nice to share some insights regarding the recommended viewing conditions.
2: Caricaturization of celebrities: I am curious about the intersection of libel/sl&er & copyright laws when you caricaturize a face belonging to a public figure or even an academic. I saw caricatures galore in this issue & was wondering if you could share some nuances &/or references that enshrine the artistic license on legal grounds. This is not so much of a critique as it is a question. Specifically, are there geographical tracts in US & beyond where such artforms can be subjected to defamation lawsuites?
3: Target audience: I did grapple with the 'whyness' of this submission. Or,  simply out, " If this comic is the answer, what is the question?". One rather reasonable take is that this work caters to an audience out there who are intrigued by AI but are outside of the proverbial "belly of the beast" who might be getting "mixed signals" from the community. In this framing, this comic renders a yeoman service in terms of setting the record straight. My gut feeling is that the techbro-model-rights-activists crowd might not be swayed by the artistic chutzpah on display here, & perhaps, such a pursuit might well be futile at inception. Any insights on who your target audience was when you created this?
4:Please consider open-sourcing the tools/code used

**Score:**

Accept: The reviewer believes the submission provides a novel and reliable scheme to improve science communication but needs improvement.

---

### Meta-Review · Program_Chairs · 2021-03-31

**Recommendation:** Accept
**Confidence:** 4

**Metareview:**

I think both the reviewers gave excellent comments, appreciation and feedback. The depth of the reviews surpasses my own understanding of the field and I would commend both the reviewers for the same.

I agree with most of the things mentioned in both reviews. I agree with R1on the fact that while this is new in ML (dissemination through comics), it has been around in other fields and the submission would greatly benefit (to the readers as well) by including a short but relevant literature review at the end. I also agree with the high-level discussion point as it would provide perspective to people who might not consider this as a "required" medium. The comics and the topic being dealt with are very interesting and intriguing to me (a person within the community) but the length might make people refer back, maybe given the digital medium, we can have some backward references to help with this over time.

R2's review is fun to read and also highlights the details I might myself have mentioned about colors, but again it looks like the authors are more experienced than us in this. The one place I felt the color scheme was hard to parse was the references page with the black font being encompassed by darker colors.

One common point in both reviews and even I feel is the legal aspect and again we aren't the experts, but a short write-up on such things (given their potential for virality) would help researchers in taking such steps in the future.

Overall, a fun read, great topic for the presentation. We would like this to be part of the workshop and become mainstream in the future. Please do incorporate the feedback from the reviewers, both of them have expertise and enthusiasm towards this work.

---

### Decision · Program_Chairs · 2021-04-01

Accept (Poster)